# A Computationally Efficient Algorithm for
# Infinite-Horizon Average-Reward Linear MDPs

**Kihyuk Hong** [1]  **Ambuj Tewari** [1]

## Abstract

We study reinforcement learning in infinite-horizon average-reward settings with linear MDPs. Previous work addresses this problem by approximating the average-reward setting by discounted setting and employing a value iteration-based algorithm that uses clipping to constrain the span of the value function for improved statistical efficiency. However, the clipping procedure requires computing the minimum of the value function over the entire state space, which is prohibitive since the state space in linear MDP setting can be large or even infinite. In this paper, we introduce a value iteration method with efficient clipping operation that only requires computing the minimum of value functions over the set of states visited by the algorithm. Our algorithm enjoys the same regret bound as the previous work while being computationally efficient, with computational complexity that is independent of the size of the state space.

## 1. Introduction

Reinforcement learning (RL) aims to learn optimal actions for an agent by interacting with the environment. Among the various RL settings, the infinite-horizon setting is particularly well-suited for applications where optimizing long-term performance is the primary objective. Examples include production system management (Yang et al., 2021; Gosavi, 2004), inventory management (Gijsbrechts et al., 2022; Giannoccaro & Pontrandolfo, 2002) and network routing (Mammeri, 2019), where interactions between the agent and the environment continue indefinitely, and the natural goal is to optimize long-term rewards.

In the infinite-horizon framework, there are two widely-used

definitions of long-term rewards. The first is the infinite-horizon discounted setting, where the objective is to maximize the discounted cumulative sum of rewards, with exponentially decaying weight assigned to future rewards. The second is the infinite-horizon average-reward setting, where the objective is to maximize the undiscounted long-term average of rewards, assigning uniform weight to future and present rewards. Learning in the average-reward setting is more challenging because its Bellman operator is not a contraction, and the widely used value iteration algorithm may fail when the transition probability model used for value iteration is not well-behaved. This complicates algorithm design, especially when the underlying transition probability model is unknown and must be estimated.

Seminal work by Auer et al. (2008) introduces a value iteration based algorithm for the infinite-horizon average-reward setting in the tabular case, where the state space and the action space are finite. To address sensitivity of the value iteration algorithm to the transition probability model, they maintain a confidence set that captures the true, well-behaved transition probability model. Their algorithm employs an extended value iteration approach, which optimally selects the transition probability model from the confidence set at each iteration. This extended value iteration method has since been extensively used in the tabular setting (Bartlett & Tewari, 2009; Fruit et al., 2018; Zhang & Ji, 2019). Beyond the tabular case, the method has also been adapted to the linear mixture MDP setting (Modi et al., 2020; Ayoub et al., 2020), where the transition probability model has a low-dimensional structure (Ayoub et al., 2020; Wu et al., 2022; Chae et al., 2025).

To our knowledge, the extended value iteration method is limited to tabular and linear mixture MDPs, as it relies on sample-efficient transition probability estimation, which is infeasible for settings like linear MDPs with large state spaces (Jin et al., 2020). In response to these limitations, researchers have explored alternative approaches for such settings. For example, Wei et al. (2021) propose a reduction to the finite-horizon episodic setting by dividing the time steps into episodes of a fixed length. This approach achieves a regret bound of $\widetilde{\mathcal{O}}(T^{3/4})$, which is suboptimal, where $T$ denotes the number of time steps. They also introduce a

[1]Department of Statistics, University of Michigan, Ann Arbor, US. Correspondence to: Ambuj Tewari <tewaria@umich.edu>.

*Proceedings of the 42nd International Conference on Machine Learning*, Vancouver, Canada. PMLR 267, 2025. Copyright 2025 by the author(s).

policy-based algorithm that alternates between policy evaluation and policy improvement steps to directly optimize the policy. This approach achieves an order-optimal regret bound of $\widetilde{\mathcal{O}}(\sqrt{T})$, but it requires a strong ergodicity assumption on the transition probability model for sample-efficient policy evaluation. Lastly, they propose another approach that achieves an order-optimal regret bound by directly solving the Bellman optimality equation as a fixed point problem, bypassing the need for value iteration. However, the fixed point problem is computationally intractable.

Another line of work on infinite-horizon average-reward RL uses a reduction to the discounted setting to leverage value iteration-based algorithms. To our knowledge, Wei et al. (2020) were the first to introduce such a method. They propose a Q-learning-based algorithm for the tabular setting that solves the discounted setting problem as a surrogate for the average-reward problem, achieving a regret bound of $\widetilde{\mathcal{O}}(T^{2/3})$. More recently, Hong et al. (2025) propose a value iteration based algorithm that clips the value function to constrain its span for statistical efficiency, achieving an order-optimal regret bound of $\widetilde{\mathcal{O}}(\sqrt{T})$. Their algorithm runs value iteration to generate a sequence of value functions to plan for the remaining time steps, and takes actions greedy with respect to the value functions until a certain information criterion of the collected trajectories doubles. Although the algorithm runs in polynomial time with respect to problem parameters, its computational complexity depends on the size of the state space. The dependency is undesirable in the linear MDP setting where the state space can be arbitrarily large. An open question arising from this line of work is:

> *Does there exist an algorithm for infinite-horizon average-reward linear MDPs with computational complexity polynomial in the problem parameters, yet independent of the size of the state space?*

In this paper, we answer the question in the affirmative by proposing an algorithm based on the following novel techniques.

**Efficient Clipping** We develop an efficient value function clipping strategy that requires the minimum of the value function to be evaluated only over the set of states visited by the algorithm, rather than the entire state space.

**Deviation-Controlled Value Iteration** We introduce a novel value iteration scheme that controls the deviation between sequences of value functions generated by value iterations with different clipping thresholds.

### 1.1. Related Work

Table 1 compares our work with previous approaches for infinite-horizon average-reward linear MDPs. FOPO solves the Bellman optimality equation directly as a fixed-point problem, which is computationally intractable, with brute-force solution requiring computational complexity that scales with $T^d$, where $d$ is the dimension of the feature representation. OLSVI.FH reduces the problem to the finite-horizon episodic setting. This approach is computationally efficient, but has suboptimal regret bound. LOOP generalizes FOPO to the general function approximation setting, but inherits the computational complexity that scales with $T^d$ for solving a fixed-point problem. MDP-EXP2 directly optimizes for the policy by alternating between policy evaluation and policy improvement. This approach is computationally efficient and achieves an order-optimal regret bound, but requires a strong assumption that all policies induce Markov chains that have uniformly bounded mixing time. $\gamma$-LSCVI-UCB reduces the average-reward problem to the discounted problem and achieves an order-optimal regret bound. However, its computational complexity scales with the size of the state space $S$. Our work is the first computationally efficient algorithm to achieve $\widetilde{\mathcal{O}}(\sqrt{T})$ regret without making strong assumptions.

**Approximation by discounted setting** The method of approximating the average-reward setting by the discounted setting has been used in various settings. It is used in the problem of finding a nearly optimal policy given access to a simulator in the tabular setting by Jin & Sidford (2021); Wang et al. (2022); Zurek & Chen (2023); Wang et al. (2023). It is also used in the online RL setting with tabular MDPs: Wei et al. (2020) propose a Q-learning based algorithm, but has $\widetilde{\mathcal{O}}(T^{2/3})$ regret. Zhang & Xie (2023) improve the regret to $\widetilde{\mathcal{O}}(\sqrt{T})$ by making use of an estimate for the span of optimal bias function. The reduction is also used in the linear mixture MDP setting by Chae et al. (2025).

**Span-constraining methods** Learning in the infinite-horizon average-reward setting requires an assumption that ensures the agent can recover from a bad state, leading to a bounded span of the optimal value function. For statistical efficiency, previous work makes use of this fact by constraining the span of the value function estimates. Bartlett & Tewari (2009) modify the extended value iteration algorithm by Auer et al. (2008) to constrain the confidence set on the model so that the spans of the models in the set are bounded. Fruit et al. (2018) propose a computationally efficient version of the algorithm proposed by Bartlett & Tewari (2009). Zhang & Ji (2019) improve the algorithm proposed by Bartlett & Tewari (2009) by constructing tighter confidence sets using a method for directly estimating the bias function. Zhang & Xie (2023) study a Q-learning-based algorithm that projects the value function to a function class of span-constrained functions. Hong et al. (2025) and Chae et al. (2025) propose a value iteration-based algorithm and clips the value function to constrain its span.

*Table 1.* Comparison of algorithms for infinite-horizon average-reward linear MDP

| Algorithm | Regret $\widetilde{\mathcal{O}}(\cdot)$ | Assumption | Computation poly$(\cdot)$ |
|---|---|---|---|
| FOPO (Wei et al., 2021) | $\mathrm{sp}(v^*)\sqrt{d^3T}$ | Bellman optimality equation | $T^d, A, d$ |
| OLSVI.FH (Wei et al., 2021) | $\sqrt{\mathrm{sp}(v^*)}(dT)^{\frac{3}{4}}$ | Bellman optimality equation | $T, A, d$ |
| LOOP (He et al., 2024) | $\sqrt{\mathrm{sp}(v^*)^3 d^3 T}$ | Bellman optimality equation | $T^d, A, d$ |
| MDP-EXP2 (Wei et al., 2021) | $d\sqrt{t_{\mathrm{mix}}^3 T}$ | Uniform Mixing | $T, A, d$ |
| $\gamma$-LSCVI-UCB (Hong et al., 2025) | $\mathrm{sp}(v^*)\sqrt{d^3T}$ | Bellman optimality equation | $T, S, A, d$ |
| $\gamma$-**DC-LSCVI-UCB (Ours)** | $\mathrm{sp}(v^*)\sqrt{d^3T}$ | Bellman optimality equation | $T, A, d$ |
| Lower Bound (Wu et al., 2022) | $\Omega(d\sqrt{\mathrm{sp}(v^*)T})$ | | |

## 2. Preliminaries

**Notations** Let $\|\boldsymbol{x}\|_A = \sqrt{x^T A x}$ for $\boldsymbol{x} \in \mathbb{R}^d$ and a positive semi-definite matrix $A \in \mathbb{R}^{d \times d}$. Let $a \vee b = \max\{a, b\}$ and $a \wedge b = \min\{a, b\}$. Let $\Delta(\mathcal{X})$ be the set of probability measures on $\mathcal{X}$. Let $[n] = \{1, \ldots, n\}$ and $[m : n] = \{m, m+1, \ldots, n\}$. Let $\mathrm{sp}(v) = \max_{s,s'} |v(s) - v(s')|$. Let $\mathrm{CLIP}(x; L, U) = (x \vee L) \wedge U$.

### 2.1. Infinite-Horizon Average-Reward RL

In this section, we formulate the infinite-horizon average-reward RL setting. We pose the RL problem as a Markov decision process (MDP) $\mathcal{M} = (\mathcal{S}, \mathcal{A}, P, r)$ where $\mathcal{S}$ is the state space, $\mathcal{A}$ is the action space, $P : \mathcal{S} \times \mathcal{A} \to \Delta(\mathcal{S})$ is the probability transition kernel and $r : \mathcal{S} \times \mathcal{A} \to \mathbb{R}$ is the reward function. We assume that rewards are bounded in $[0, 1]$, a standard and mild assumption that can be enforced by rescaling. We assume $\mathcal{S}$ is a measurable space with possibly infinite number of elements and $\mathcal{A}$ is a finite set. The deterministic reward function $r$ is known to the learner while the probability transition kernel $P$ is unknown to the learner.

The interaction protocol between the learner and the MDP is as follows. The environment first reveals the starting state $s_1 \in \mathcal{S}$ to the learner. Then, at each time step $t = 1, 2, \ldots$, the learner chooses an action $a_t \in \mathcal{A}$ and receives the reward $r(s_t, a_t)$. The environment transitions to the next state $s_{t+1}$ sampled from $P(\cdot|s_t, a_t)$.

In the infinite-horizon average-reward setting, the performance of a policy is evaluated using the long-term average reward. Consider a stationary policy $\pi : \mathcal{S} \to \Delta(\mathcal{A})$ where $\pi(a|s)$ denotes the probability of choosing action $a$ in state $s$. The average reward of policy $\pi$ starting from an initial state $s$ is defined as

$$J^\pi(s) \coloneqq \liminf_{T \to \infty} \frac{1}{T} \mathbb{E}^\pi \left[ \sum_{t=1}^T r(s_t, a_t) \,\bigg|\, s_1 = s \right]$$

where the expectation $\mathbb{E}^\pi[\cdot]$ is taken over the probability distribution on the trajectory $(s_1, a_1, s_2, a_2, \ldots)$ induced by the interaction between $P$ and $\pi$.

The performance of an algorithm interacting with the environment over $T$ steps is evaluated through its regret relative to the best stationary policy $\pi^*$ that maximizes $J^\pi(s_1)$. Writing $J^*(s_1) \coloneqq J^{\pi^*}(s_1)$, the regret after $T$ steps is defined as

$$R_T \coloneqq \sum_{t=1}^T (J^*(s_1) - r(s_t, a_t)).$$

The interaction protocol for the infinite-horizon setting, unlike the interaction protocol for the finite-horizon episodic setting, the state is never reset. Consequently, if the agent enters a bad state with low future reward and recovering from the bad state and reaching a good state is impossible, then the agent becomes trapped in the bad state and suffers regret linear in the number of remaining time steps. As discussed by Bartlett & Tewari (2009), an additional assumption on the structure of the MDP is required to avoid the pathological case. At the very least, we want the gain $J^*(s)$ to be constant: $J^*(s) = J^*$ for all $s \in \mathcal{S}$. This implies no matter what the current state is, following the optimal policy $\pi^*$ attains the optimal long-term average reward $J^*$, precluding the case of getting trapped in a bad state. We follow Wei et al. (2021) and make the following structural assumption on the MDP.

**Assumption A** (Bellman optimality equation)**.** There exist $J^* \in \mathbb{R}$ and functions $v^* : \mathcal{S} \to \mathbb{R}$ and $q^* : \mathcal{S} \times \mathcal{A} \to \mathbb{R}$ such that for all $(s, a) \in \mathcal{S} \times \mathcal{A}$, we have

$$J^* + q^*(s, a) = r(s, a) + [Pv^*](s, a)$$
$$v^*(s) = \max_{a \in \mathcal{A}} q^*(s, a).$$

As shown by Wei et al. (2021), a tuple $(J^*, q^*, v^*)$ that satisfies the equations in the assumption above has the following properties. The policy $\pi^*$ that deterministically selects an action from $\mathrm{argmax}_a q^*(s, a)$ at each state $s \in \mathcal{S}$ is an optimal policy. Moreover, such $\pi^*$ always gives an optimal average reward $J^{\pi^*}(s) = J^*$ for all initial states $s \in \mathcal{S}$. Since the optimal average reward is independent of the ini-

tial state, we can simply write the regret as

$$R_T = \sum_{t=1}^{T} (J^* - r(s_t, a_t)).$$

The functions $v^*(s)$ and $q^*(s,a)$ have the interpretation of the relative advantage of starting with $s$ and $(s,a)$, respectively, and are called bias functions. A pair of functions $(v^*, q^*)$ that satisfies the Bellman optimality equation is $v^*(s) = \lim_{N\to\infty} \mathbb{E}^{\pi^*}[\sum_{t=1}^{N} r(s_t, a_t) - J^* | s_1 = s]$ and $q^*(s,a) = \lim_{N\to\infty} \mathbb{E}^{\pi^*}[\sum_{t=1}^{N} r(s_t, a_t) - J^* | s_1 = s, a_1 = a]$.

**Remark 2.1.** The Bellman optimality equation assumption is weaker than the weakly communicating assumption, which states that the state space can be partitioned into a set of transient states, which the agent never revisits once it leaves, and a set of recurrent states, where the agent can reach any state from any other under some policy. In turn, the weakly communicating assumption is weaker than the ergodic assumption, which requires that for every policy, the induced Markov chain is irreducible and aperiodic.

The span of the bias, $\mathrm{sp}(v^*) = \max_{s,s'\in\mathcal{S}} v^*(s) - v^*(s')$, quantifies the worst-case difference in value between any two states. Intuitively, entering a suboptimal state incurs regret that scales with $\mathrm{sp}(v^*)$, suggesting that problems with large $\mathrm{sp}(v^*)$ are more challenging to learn. Following previous work (Bartlett & Tewari, 2009; Wei et al., 2020), we assume $\mathrm{sp}(v^*)$ is known to the learner. This assumption can be relaxed by instead assuming access to an upper bound on $\mathrm{sp}(v^*)$, but in this case, the regret of our proposed algorithm will scale with the upper bound.

**Remark 2.2.** Whether sample-efficient learning is possible without knowing the span in advance has been an open question for a long time, and many existing works rely on this assumption. A recent result by Boone & Zhang (2024) shows for the first time that it can be avoided in the tabular setting. However, extending their technique to the linear MDP setting remains a significant challenge and likely require a major breakthrough. We leave this to future work.

## 2.2. Infinite-Horizon Discounted Setting

The key algorithm design employed in this paper is to approximate the infinite-horizon average-reward setting by the infinite-horizon discounted setting with a discounting factor $\gamma \in [0,1)$ chosen by the learner. Under the discounted setting, the performance measure is the discounted sum of rewards $\sum_{t=1}^{\infty} \gamma^{t-1} r(s_t, a_t)$. When normalized by a factor $(1-\gamma)$, the resulting normalized discounted sum is a weighted average of the reward sequence $r(s_1, a_1), r(s_2, a_2), \ldots$. The decay rate of the weight sequence is governed by the discounting factor $\gamma$. As $\gamma$ approaches 1, the decay becomes slower and the normalized

discounted sum should approach average of the reward sequence. To make this intuition precise, we first define value functions for a policy $\pi$ under the discounted setting by

$$V_\gamma^\pi(s) = \mathbb{E}^\pi \left[ \sum_{t=1}^{\infty} \gamma^{t-1} r(s_t, a_t) | s_1 = s \right]$$

$$Q_\gamma^\pi(s,a) = \mathbb{E}^\pi \left[ \sum_{t=1}^{\infty} \gamma^{t-1} r(s_t, a_t) | s_1 = s, a_1 = a \right].$$

We write the optimal value functions under the discounted setting as

$$V_\gamma^*(s) = \max_\pi V^\pi(s), \quad Q_\gamma^*(s,a) = \max_\pi Q_\gamma^\pi(s,a).$$

Previous informal discussion suggests that the normalized value function $(1-\gamma)V_\gamma^*(s)$ to be close to the gain under the average-reward setting $J^*$. The following lemma makes the relation between the infinite-horizon average-reward setting and the discounted setting formal.

**Lemma 2.3** (Lemma 2 in Wei et al. (2020))**.** *For any $\gamma \in [0,1)$, the optimal value function $V^*$ for the infinite-horizon discounted setting with discounting factor $\gamma$ satisfies*

*(i) $sp(V_\gamma^*) \le 2sp(v^*)$ and*

*(ii) $|(1-\gamma)V_\gamma^*(s) - J^*| \le (1-\gamma)sp(v^*)$ for all $s \in \mathcal{S}$.*

The lemma above suggests that the difference between the optimal average reward $J^*$ and the optimal discounted cumulative reward normalized by the factor $(1-\gamma)$ is small as long as $\gamma$ is close to 1. Hence, we can expect the policy optimal under the discounted setting will be nearly optimal for the average-reward setting, provided $\gamma$ is sufficiently close to 1.

## 2.3. Linear MDPs

The linear MDP setting is a widely-studied setting in RL theory literature that allows sample efficient learning in large state space by assuming a low-dimensional feature representation of the state-action pair. This representation allows for generalization to unseen states, yielding sample complexity that is independent of the size of the state space. The linear MDP model imposes the following structural assumptions on the MDP:

**Assumption B** (Linear MDP (Jin et al., 2020))**.** We assume that the transition and the reward functions can be expressed as a linear function of a known $d$-dimensional feature map $\varphi : \mathcal{S} \times \mathcal{A} \to \mathbb{R}^d$ such that for any $(s,a) \in \mathcal{S} \times \mathcal{A}$, we have

$$r(s,a) = \langle \varphi(s,a), \theta \rangle, \quad P(s'|s,a) = \langle \varphi(s,a), \mu(s') \rangle$$

where $\mu(s') = (\mu_1(s'), \ldots, \mu_d(s'))$ for $s' \in \mathcal{S}$ is a vector of $d$ unknown measures on $\mathcal{S}$ and $\theta \in \mathbb{R}^d$ is a known parameter for the reward function.

we further assume, without loss of generality, the following boundedness conditions:

$$\|\boldsymbol{\varphi}(s,a)\|_2 \le 1 \text{ for all } (s,a) \in \mathcal{S} \times \mathcal{A},$$
$$\|\boldsymbol{\theta}\|_2 \le \sqrt{d}, \quad \|\boldsymbol{\mu}(\mathcal{S})\|_2 \le \sqrt{d}. \tag{1}$$

Such a boundedness assumption is commonly made, without loss of generality (Wei et al., 2021), when studying the linear MDP setting.

**Remark 2.4.** Wei et al. (2021) show that the boundedness assumption can be made without loss of generality with the following reasoning. Given a parameterization $\boldsymbol{\theta}$ and $\boldsymbol{\varphi}$ for the reward function, we can rescale $\boldsymbol{\theta}$ and $\boldsymbol{\varphi}$ such that $\|\boldsymbol{\varphi}(\cdot,\cdot)\|_2 \le 1$. Then, they show there exists an invertible transformation $A : \mathbb{R}^d \to \mathbb{R}^d$ such that the minimum volume enclosing ellipsoid (MVEE) of $A\Phi \cup (-A\Phi)$ is a unit ball. A transformed feature mapping $\varphi'(s,a) = A\varphi(s,a)$ and a transformed parameter $\theta' = A^{-1}\theta$ leads to the same reward function with $\|\theta'\|_2 \le 1$, as desired. As discussed by Wei et al. (2021), this transformation depends only on the feature mapping $\varphi$, not on the parameter $\theta$, and can thus be applied during the feature design stage As shown in Theorem 8 of Hazan & Karnin (2016), computing such a transformation takes time $\mathcal{O}((\sqrt{SA}+d)S^3A^3)$.

As discussed by Jin et al. (2020), although the transition model $P$ is linear in the $d$-dimensional feature mapping $\varphi$, $P$ still has $|\mathcal{S}|$ degrees of freedom as the measure $\boldsymbol{\mu}$ is unknown, making the estimation of the model $P$ difficult. For sample efficient learning, we rely on the fact that $[PV](s,a)$ is linear in $\varphi(s,a)$ for any function $V : \mathcal{S} \to \mathbb{R}$ so that $[PV](s,a) = \langle \boldsymbol{\varphi}(s,a), \boldsymbol{w}^*(V) \rangle$ where $\boldsymbol{w}^*(V) := \int_{s' \in \mathcal{S}} V(s') \boldsymbol{\mu}(ds')$. Indeed,

$$
\begin{aligned}
[PV](s,a) &:= \int_{s' \in \mathcal{S}} V(s') P(ds'|s,a) \\
&= \int_{s' \in \mathcal{S}} V(s') \langle \boldsymbol{\varphi}(s,a), \boldsymbol{\mu}(ds') \rangle \\
&= \langle \boldsymbol{\varphi}(s,a), \int_{s' \in \mathcal{S}} V(s') \boldsymbol{\mu}(ds') \rangle.
\end{aligned}
$$

Exploiting the linearity, we can estimate $\boldsymbol{w}^*(V)$ given a trajectory data $(s_1, a_1, \ldots, s_{t-1}, a_{t-1}, s_t)$ via linear regression as follows:

$$\widehat{\boldsymbol{w}}_t(V) := \Lambda_t^{-1} \sum_{\tau=1}^{t-1} V(s_{\tau+1}) \cdot \boldsymbol{\varphi}(s_\tau, a_\tau)$$

where $\Lambda_t = \lambda I + \sum_{\tau=1}^{t-1} \boldsymbol{\varphi}(s_t, a_t)\boldsymbol{\varphi}(s_t, a_t)^\top$. With such a regression coefficient, we estimate $[PV](s,a)$ by

$$[\widehat{P}_t V](s,a) := \langle \boldsymbol{\varphi}(s,a), \widehat{\boldsymbol{w}}_t(V - V(s_1)) \rangle + V(s_1).$$

We estimate $[PV](s,a)$ by estimating $[P(V - V(s_1))](s,a)$ and then adding back $V(s_1)$. This allows bounding the norm of the regression coefficient $\|\widehat{\boldsymbol{w}}_t(V - V(s_1))\|_2$ by a bound that scales with the span of $V$ instead of the magnitude of $V$, which is required for getting a sharp regret bound. A similar technique is used by Hong et al. (2025).

---

**Algorithm 1** $\gamma$-LSCVI-UCB (Hong et al., 2025)

**Input:** Discounting factor $\gamma \in [0,1)$, regularization constant $\lambda > 0$, span $H > 0$, bonus factor $\beta > 0$.
**Initialize:** $k \leftarrow 1, t_k \leftarrow 1, \Lambda_1 \leftarrow \lambda I, Q_t^1(\cdot,\cdot) \leftarrow \frac{1}{1-\gamma}$ for $t \in [T]$.
1: Receive state $s_1$.
2: **for** time step $t = 1, \ldots, T$ **do**
3:     Take action $a_t = \operatorname{argmax}_a Q_t^t(s_t, a)$.
4:     Receive reward $r(s_t, a_t)$; Receive next state $s_{t+1}$.
5:     $\Lambda_t \leftarrow \Lambda_{t-1} + \boldsymbol{\varphi}(s_t, a_t)\boldsymbol{\varphi}(s_t, a_t)^\top$.
6:     **if** $2\det(\Lambda_{t_k}) < \det(\Lambda_t)$ **then**
7:         $k \leftarrow k+1, t_k \leftarrow t+1$.
8:         $V_{T+1}^{t+1}(\cdot) \leftarrow \frac{1}{1-\gamma}$.
9:         **for** $u = T, T-1, \ldots, t_k$ **do**
10:           $Q_u^{t+1}(\cdot,\cdot) \leftarrow \Big( r(\cdot,\cdot) + \gamma([\widehat{P}_{t_k} V_{u+1}^{t+1}](\cdot,\cdot)$
                               $+ \beta\|\boldsymbol{\varphi}(\cdot,\cdot)\|_{\Lambda_{t_k}^{-1}})\Big) \wedge \frac{1}{1-\gamma}$.
11:           $\widetilde{V}_u^{t+1}(\cdot) \leftarrow \max_a Q_u^{t+1}(\cdot,a)$.
12:           $V_u^{t+1}(\cdot) \leftarrow \text{CLIP}(\widetilde{V}_u^{t+1}(\cdot);$
             $\min_{s' \in \mathcal{S}} \widetilde{V}_u^{t+1}(s'), \min_{s' \in \mathcal{S}} \widetilde{V}_u^{t+1}(s') + H)$.
13:         **end for**
14:     **else**
15:         $Q_u^{t+1} \leftarrow Q_u^t, V_u^{t+1} \leftarrow V_u^t$ for all $u \in [t+1 : T]$.
16:     **end if**
17: **end for**

---

### 2.4. Previous Work

In this section, we review the closely related work of Hong et al. (2025) to highlight the contributions of our paper. They propose an algorithm, $\gamma$-LSCVI-UCB (Algorithm 1), which is an optimistic value iteration based algorithm for infinite-horizon average-reward linear MDPs. At time step $t$, a sequence of value functions $Q_T^t, Q_{T-1}^t, \ldots, Q_t^t$ is computed by running value iterations (Line 8-13) to plan for the best action at time $t$, considering the number of time steps remaining. In the next time step $t+1$, instead of running value iteration again to incorporate new transition data observed at time step $t$, the algorithm reuses the value function $Q_{t+1}^t$ generated previously. Value iteration is only rerun when the determinant of the covariance matrix $\Lambda_t = \lambda I + \sum_{\tau=1}^t \boldsymbol{\varphi}(s_t, a_t)\boldsymbol{\varphi}(s_t, a_t)^\top$ doubles (Line 6).

**Clipped Value Iteration** A key ingredient of the algorithm is the value clipping step, which constrains the span of the value function estimate to improve statistical efficiency. The optimal value function $V_\gamma^*$ under the discounted setting has a span bounded by $2 \cdot \text{sp}(v^*)$ (Lemma 2.3), which implies the range $V_\gamma^*$ is contained in the interval $[\min_{s \in \mathcal{S}} V_\gamma^*(s), \min_{s \in \mathcal{S}} V_\gamma^*(s) + 2 \cdot \text{sp}(v^*)]$. Building on this fact, the algorithm clips the optimistic value function estimate $\widetilde{V}$ to the interval $[\min_{s \in \mathcal{S}} \widetilde{V}(s), \min_{s \in \mathcal{S}} \widetilde{V}(s) + H]$ to constrain its span (Line 7). We refer to the lower bound

of this interval of the clipping operation as *clipping threshold*. The clipping ensures that the concentration bound for the estimate $[\widehat{P}V](\cdot, \cdot)$ scales with $\mathrm{sp}(v^*)$, rather than $\frac{1}{1-\gamma}$, which is crucial for obtaining a tight regret bound.

**Key Step of Regret Analysis**   In their regret analysis, one of the terms in the regret decomposition is

$$\sum_{t=1}^{T} V_{t+1}^t(s_{t+1}) - \widetilde{V}_{t+1}^{t+1}(s_{t+1}).$$

This term can be bounded using the fact that $V_{t+1}^{t+1}(s_{t+1}) \leq \widetilde{V}_{t+1}^{t+1}(s_{t+1})$, and that $V_{t+1}^t(s_{t+1}) = V_{t+1}^{t+1}(s_{t+1})$ whenever the same sequence of value functions is used for the time steps $t$ and $t+1$. Since the sequence of value functions is only updated when the covariance matrix $\Lambda_t$ doubles, which can be shown to happen only $\mathcal{O}(d \log T)$ times, we can get a tight regret bound.

**Computational Complexity**   However, their clipping step (Line 7) requires taking the minimum of the value function estimate $\widetilde{V}(\cdot)$ over the entire state space $\mathcal{S}$, leading to computational complexity linear in the size of the state space, which can be prohibitive when the state space is large or infinite. The main contribution of our paper addresses this issue by designing an algorithm that only takes the minimum over the states that have been visited by the learner, removing the dependency of the size of the state space on the computational complexity. As discussed in the next section, additional algorithmic trick is required for controlling the deviation of sequences of value functions generated under different clipping thresholds.

## 3. Algorithm Design and Analysis

In this section, we present our algorithm, *discounted Deviation Controlled Least Squares Clipped Value Iteration with Upper Confidence Bound* ($\gamma$-DC-LSCVI-UCB, Algorithm 2), which improves computational complexity of the previous algorithm. The part of the proposed algorithm that enables computational efficiency is highlighted in red.

### 3.1. Computationally Efficient Clipping

The algorithm design is centered around bounding the term

$$\sum_{t=1}^{T-1} V_{t+1}^t(s_{t+1}) - \widetilde{V}_{t+1}^{t+1}(s_{t+1}),$$

where $\{\widetilde{V}_u^t\}_{u \in [t:T]}$ is the sequence of value functions generated at time step $t$, and $\{V_u^t\}_{u \in [t:T]}$ is the sequence of clipped value functions generated at time step $t$. Note that the clipped value function $V_{t+1}^t$ in the summation is generated at time step $t$, prior to observing the next state $s_{t+1}$.

With unlimited compute power, the $\gamma$-LSCVI-UCB algorithm by previous work uses $\min_{s \in \mathcal{S}} \widetilde{V}_{t+1}^t(s)$ as the clipping threshold, which allows bounding $V_{t+1}^t$ evaluated at $s_{t+1}$ by

$$V_{t+1}^t(s_{t+1})$$
$$= \mathrm{CLIP}(\widetilde{V}_{t+1}^t(s_{t+1}); \min_{s \in \mathcal{S}} \widetilde{V}_{t+1}^t(s), \min_{s \in \mathcal{S}} \widetilde{V}_{t+1}^t(s) + H)$$
$$\leq \widetilde{V}_{t+1}^t(s_{t+1})$$

where the inequality only holds because $\min_{s \in \mathcal{S}} \widetilde{V}_{t+1}^t(s) \leq \widetilde{V}_{t+1}^t(s_{t+1})$. The algorithm $\gamma$-LSCVI-UCB also reuses the sequence of value functions most of the time steps, such that $\widetilde{V}_{t+1}^t(s_{t+1}) = \widetilde{V}_{t+1}^{t+1}(s_{t+1})$, allowing the bound $V_{t+1}^t(s_{t+1}) - \widetilde{V}_{t+1}^{t+1}(s_{t+1}) \leq 0$.

For computational efficiency, suppose we use $m_t$ as the clipping threshold instead of $\min_{s \in \mathcal{S}} \widetilde{V}_{t+1}^t(s)$, where $m_t$ is computed using states $s_1, \ldots, s_t$ only. Then, the bound $V_{t+1}^t(s_{t+1}) \leq \widetilde{V}_{t+1}^t(s_{t+1})$ may no longer hold because

$$V_{t+1}^t(s_{t+1}) = \mathrm{CLIP}(\widetilde{V}_{t+1}^t(s_{t+1}); m_t, m_t + H) \geq m_t$$

and we may have $m_t > \widetilde{V}_{t+1}^t(s_{t+1})$ since we cannot look ahead $s_{t+1}$ when choosing the clipping threshold $m_t$. We can instead get a bound with an error term:

$$V_{t+1}^t(s_{t+1}) = \mathrm{CLIP}(\widetilde{V}_{t+1}^t(s_{t+1}); m_t, m_t + H)$$
$$\leq \widetilde{V}_{t+1}^t(s_{t+1}) + \max\{m_t - \widetilde{V}_{t+1}^t(s_{t+1}), 0\}.$$

One key idea of handling the sum of the error terms is to choose $m_{t+1} = \widetilde{V}_{t+1}^t(s_{t+1}) \wedge m_t$ (Line 15), leading to

$$V_{t+1}^t(s_{t+1}) \leq \widetilde{V}_{t+1}^t(s_{t+1}) + \Delta_t$$

where $\Delta_t = m_t - m_{t+1}$. Then the sum of the errors $\Delta_t$ can then be bounded using a telescoping sum.

The clipping threshold $m_{t+1} = \widetilde{V}_{t+1}^t(s_{t+1}) \wedge m_t$ may change every time step. Hence, after advancing to the next time step $t+1$ and computing the new threshold $m_{t+1}$, the algorithm computes $Q_{t+1}^{t+1}$ afresh, which involves generating a sequence of value functions $V_T^{t+1}, \ldots, V_{t+1}^{t+1}$ by running clipped value iteration with the new threshold $m_{t+1}$. Therefore, unlike previous work that ensures $\widetilde{V}_{t+1}^t(s_{t+1}) = \widetilde{V}_{t+1}^{t+1}(s_{t+1})$ by reusing the sequence of value functions, we need to control the difference between $\widetilde{V}_{t+1}^t(s_{t+1})$ and $\widetilde{V}_{t+1}^{t+1}(s_{t+1})$ to be able to bound

$$V_{t+1}^t(s_{t+1}) \leq \widetilde{V}_{t+1}^t(s_{t+1}) + \Delta_t \approx \widetilde{V}_{t+1}^{t+1}(s_{t+1}) + \Delta_t.$$

The next section discusses the algorithm design for ensuring $\widetilde{V}_{t+1}^t \approx \widetilde{V}_{t+1}^{t+1}$.

**Algorithm 2** $\gamma$-DC-LSCVI-UCB
___

**Input:** Discounting factor $\gamma \in [0, 1)$, regularization constant $\lambda > 0$, span $H > 0$, bonus factor $\beta > 0$.

**Initialize:** $\Lambda_1 \leftarrow \lambda I$, $m_{-1} \leftarrow \infty$, $m_0 \leftarrow \infty$, $m_1 \leftarrow \frac{1}{1-\gamma}$,

$\quad \widetilde{Q}_u^0(\cdot, \cdot) \leftarrow \frac{1}{1-\gamma}$, $\widetilde{Q}_u^{-1}(\cdot, \cdot) \leftarrow \frac{1}{1-\gamma}$.

1: Receive state $s_1$.
2: **for** $t = 1, \ldots, T$ **do**
3: $\quad V_{T+1}^t(\cdot) \leftarrow \frac{1}{1-\gamma}$.
4: $\quad$ **for** $u = T, T-1, \ldots, t$ **do**
5: $\quad\quad \widetilde{Q}_u^t(\cdot, \cdot) \leftarrow \Big( r(\cdot, \cdot) + \gamma([\widehat{P}_t V_{u+1}^t](\cdot, \cdot)$
$\quad\quad\quad\quad\quad\quad\quad\quad + \beta \|\boldsymbol{\varphi}(\cdot, \cdot)\|_{\Lambda_t^{-1}}) \Big) \wedge \frac{1}{1-\gamma}$.
6: $\quad\quad {\color{red} U_u^t(\cdot, \cdot) \leftarrow \widetilde{Q}_u^{t-1}(\cdot, \cdot) \wedge \widetilde{Q}_u^{t-2}(\cdot, \cdot).}$
7: $\quad\quad {\color{red} L_u^t(\cdot, \cdot) \leftarrow (\widetilde{Q}_u^{t-1}(\cdot, \cdot) - m_{t-1} + m_t)}$
$\quad\quad\quad\quad {\color{red} \vee (\widetilde{Q}_u^{t-2}(\cdot, \cdot) - m_{t-2} + m_t).}$
8: $\quad\quad {\color{red} Q_u^t(\cdot, \cdot) \leftarrow \text{CLIP}(\widetilde{Q}_u^t(\cdot, \cdot); L_u^t(\cdot, \cdot), U_u^t(\cdot, \cdot)).}$
9: $\quad\quad \widetilde{V}_u^t(\cdot) \leftarrow \max_a Q_u^t(\cdot, a).$
10: $\quad\quad V_u^t(\cdot) \leftarrow \text{CLIP}(\widetilde{V}_u^t(\cdot); m_t, m_t + H).$
11: $\quad$ **end for**
12: $\quad$ Take action $a_t \leftarrow \text{argmax}_{a \in \mathcal{A}} Q_t^t(s_t, a).$
13: $\quad$ Receive reward $r(s_t, a_t)$. Receive next state $s_{t+1}$.
14: $\quad \Lambda_{t+1} \leftarrow \Lambda_t + \boldsymbol{\varphi}(s_t, a_t)\boldsymbol{\varphi}(s_t, a_t)^\top.$
15: $\quad {\color{red} m_{t+1} \leftarrow \widetilde{V}_{t+1}^t(s_{t+1}) \wedge m_t.}$
16: **end for**

### 3.2. Deviation-Controlled Value Iteration

Previous discussion suggests we need to bound the difference between sequences of value functions $\{\widetilde{V}_u^t\}_{u \in [T]}$ and $\{\widetilde{V}_u^{t+1}\}_{u \in [T]}$ generated by value iterations using different clipping thresholds $m_t$ and $m_{t+1}$. We would expect that the difference between sequences of value functions to be bounded by the difference in clipping thresholds $m_t - m_{t+1}$. Surprisingly, a naive adaptation of the previous work $\gamma$-LSCVI-UCB, fails to control the difference. To see this, consider the following clipped value iteration procedure that generates a sequence of value functions $\{\widetilde{V}_u^t\}_u$ at time step $t$ using the clipping threshold $m_t$.

$\quad V_{T+1}^t(\cdot) \leftarrow \frac{1}{1-\gamma}.$
$\quad$ **for** $u = T, T-1, \ldots, t$ **do**
$\quad\quad Q_u^t(\cdot, \cdot) \leftarrow \Big( r(\cdot, \cdot) + \gamma([\widehat{P}_t V_{u+1}^t](\cdot, \cdot)$
$\quad\quad\quad\quad\quad\quad\quad\quad + \beta \|\boldsymbol{\varphi}(\cdot, \cdot)\|_{\Lambda_t^{-1}}) \Big) \wedge \frac{1}{1-\gamma}.$
$\quad\quad \widetilde{V}_u^t(\cdot) \leftarrow \max_a Q_u^t(\cdot, a).$
$\quad\quad V_u^t(\cdot) \leftarrow \text{CLIP}(\widetilde{V}_u^t(\cdot); m_t, m_t + H).$
$\quad$ **end for**

We argue that controlling the difference $\|\widetilde{V}_{u+1}^t - \widetilde{V}_{u+1}^{t+1}\|_\infty \leq \Delta$ for $\Delta = m_t - m_{t+1}$ at value iteration index $u+1$ does not necessarily control the difference $\|\widetilde{V}_u^t - \widetilde{V}_u^{t+1}\|_\infty$ at the next value iteration. To see this, suppose $\|\widetilde{V}_{u+1}^t - \widetilde{V}_{u+1}^{t+1}\|_\infty \leq \Delta$.

Then, by value iteration, we have

$$\|\widetilde{V}_u^t - \widetilde{V}_u^{t+1}\|_\infty \leq \|Q_u^t - Q_u^{t+1}\|_\infty \approx \|\widehat{P}_t(V_{u+1}^t - V_{u+1}^{t+1})\|_\infty.$$

It is natural to expect that $\|V_{u+1}^t - V_{u+1}^{t+1}\|_\infty \leq \Delta$ would imply $\|\widehat{P}_t(V_{u+1}^t - V_{u+1}^{t+1})\|_\infty \leq \Delta$. This is true when $[\widehat{P}_t V](s, a)$ is an expectation of $V(\cdot)$ with respect to an empirical probability distribution $\widehat{P}_t(\cdot|\cdot, \cdot)$, which is the case for the tabular setting (see Appendix B.1 for more discussion). However, in the linear MDP setting, and more generally in general value function approximation setting, $[\widehat{P}_t V](s, a)$ is defined through a regression: $[\widehat{P}_t V](s, a) = \langle \boldsymbol{\varphi}(s, a), \widehat{\boldsymbol{w}}_t(V_{u+1}^t - V_{u+1}^{t+1}) \rangle$, which can be arbitrarily larger than $\Delta$ as shown in the next lemma.

**Lemma 3.1.** *There exist* $\boldsymbol{\phi}_1, \ldots, \boldsymbol{\phi}_n \in \mathbb{R}^d$ *with* $\|\boldsymbol{\phi}_i\| \leq 1$ *for* $i = 1, \ldots, n$, *and* $y_1, \ldots, y_n \in \mathbb{R}$ *with* $|y_i| \leq \Delta$, $i = 1, \ldots, n$ *for any* $\Delta > 0$, *such that*

$$|\langle \boldsymbol{w}_n, \boldsymbol{\phi} \rangle| \geq \frac{1}{2}\Delta\sqrt{n}$$

*for some* $\boldsymbol{\phi} \in \mathbb{R}^d$ *where* $\boldsymbol{w}_n$ *is the regression coefficient* $\boldsymbol{w}_n = \Lambda_n^{-1} \sum_{i=1}^n y_i \boldsymbol{\phi}_i$ *where* $\Lambda_n = \sum_{i=1}^n \boldsymbol{\phi}_i \boldsymbol{\phi}_i^\top + \lambda I$.

To address this issue, we propose a novel value iteration procedure that explicitly controls the deviation of a sequence of value functions from its previous sequences. The key idea is to clip the value function $\widetilde{Q}_u^t$ so that its values do not deviate too much from value functions $\widetilde{Q}_u^{t-1}$ and $\widetilde{Q}_u^{t-2}$ from previously generated sequences of value functions (Line 6-8). With this scheme, we can bound the difference between $\widetilde{V}_u^t$ and $\widetilde{V}_u^{t+1}$ as follows.

**Lemma 3.2.** *When running* $\gamma$-DC-LSCVI-UCB *(Algorithm 2), we have*

$$|\widetilde{V}_u^{t+1}(s) - \widetilde{V}_u^t(s)| \leq m_{t-1} - m_{t+1}$$

*for all* $t \in [T]$, $u \in [t : T]$ *and for all* $s \in \mathcal{S}$.

The lemma above says that the sequence of value functions $\{\widetilde{V}_u^{t+1}\}_{u \in [t+1:T]}$ generated at time step $t + 1$ deviates from the chain of value functions $\{\widetilde{V}_u^t\}_{u \in [t:T]}$ by at most $m_{t-1} - m_{t+1}$. This deviation control enables bounding the term $\sum_{t=1}^{T-1} V_{t+1}^t(s_{t+1}) - \widetilde{V}_{t+1}^{t+1}(s_{t+1})$, which we demonstrate in the next section.

### 3.3. Regret Analysis

In this section, we outline a regret analysis for our algorithm. Central to the regret analysis is the following concentration bound for the estimate $\widehat{P}_t V$.

**Lemma 3.3.** *With probability at least* $1 - \delta$, *there exists an absolute constant* $c_\beta$ *such that for* $\beta = c_\beta \cdot Hd\sqrt{\log(dT/\delta)}$,

$$|[\widehat{P}_t V_u^t](s, a) - [P V_u^t](s, a)| \leq \beta \|\boldsymbol{\varphi}(s, a)\|_{\Lambda_t^{-1}}$$

*for all* $t \in [T]$, $u \in [t : T]$ *and* $(s, a) \in \mathcal{S} \times \mathcal{A}$.

A proof for the lemma above first finds a concentration bound for $\widehat{P}_t V$ for a fixed value function $V : \mathcal{S} \to \mathbb{R}$ using a concentration bound for vector-valued self-normalized processes. Then, an $\epsilon$-net covering argument is used to get a uniform bound on the function class that captures all value functions $V_u^t$ encountered by the algorithm. For this to work, we require the function class to have low covering number. We can show that the log covering number of the function class that captures functions $\widetilde{Q}_u^t$ can be bounded by $\widetilde{\mathcal{O}}(d^2)$, which amounts to covering the $d \times d$ matrices $\Lambda_t$. Since $Q_u^t$ is a function of 5 functions in this function class, the log covering number of the function class that captures $Q_u^t$ is bounded by $\widetilde{\mathcal{O}}(d^2)$. With the concentration inequality, and the fact that the algorithm uses $\beta \|\varphi(s, a)\|_{\Lambda_t^{-1}}$ as the bonus term, we get the following results.

**Lemma 3.4** (Optimism). *With probability at least $1 - \delta$, for all $t \in [T]$ and $u \in [t : T]$ and $s \in \mathcal{S}$, we have*

$$V_u^t(s) \geq V^*(s),$$

*as long as the input argument $H$ is chosen such that $H \geq 2 \cdot sp(v^*)$.*

**Lemma 3.5.** *With probability at least $1 - \delta$, we have for all $t \in [4 : T]$ and $u \in [t : T]$ that*

$$\begin{aligned} Q_u^t(s, a) &\leq r(s, a) + \gamma[PV_{u+1}^t](s, a) \\ &\quad + 2\beta \|\varphi(s, a)\|_{\Lambda_t^{-1}} + 2(m_{t-3} - m_t) \end{aligned}$$

*for all $(s, a) \in \mathcal{S} \times \mathcal{A}$.*

Using the lemma above, the regret can be bounded by

$$\begin{aligned} R_T &= \sum_{t=1}^{T}(J^* - r(s_t, a_t)) \\ &\leq \sum_{t=4}^{T}(J^* - Q_t^t(s_t, a_t) + \gamma[PV_{t+1}^t](s_t, a_t) \\ &\quad + 2\beta\|\varphi(s_t, a_t)\|_{\Lambda_t^{-1}} + 2(m_{t-3} - m_t)) + \mathcal{O}(1) \end{aligned}$$

which can be decomposed into

$$\begin{aligned} &= \underbrace{\sum_{t=4}^{T}(J^* - (1-\gamma)V_{t+1}^t(s_{t+1}))}_{(a)} \\ &\quad + \underbrace{\sum_{t=4}^{T}(V_{t+1}^t(s_{t+1}) - \widetilde{V}_t^t(s_t))}_{(b)} \\ &\quad + \gamma\underbrace{\sum_{t=4}^{T}([PV_{t+1}^t](s_t, a_t) - V_{t+1}^t(s_{t+1}))}_{(c)} \\ &\quad + 2\underbrace{\beta\sum_{t=4}^{T}\|\varphi(s_t, a_t)\|_{\Lambda_t^{-1}}}_{(d)} + \mathcal{O}(\frac{1}{1-\gamma}). \end{aligned}$$

where we use $Q_t^t(s_t, a_t) = \widetilde{V}_t^t(s_t)$ by the choice of $a_t$ by the algorithm. Each term can be bounded as follows.

**Bounding (a)** By the optimism result (Lemma 3.4), we have $V_u^t(s) \geq V^*(s)$ for all $t \in [T]$ and $u \in [t : T]$ with high probability. It follows that

$$\begin{aligned} J^* - (1 - \gamma)V_{t+1}^t(s_{t+1}) &\leq J^* - (1 - \gamma)V^*(s_{t+1}) \\ &\leq (1 - \gamma)sp(v^*) \end{aligned}$$

where the last inequality is by the bound on the error of approximating the average-reward setting by the discounted setting provided in Lemma 2.3. Hence, the term $(a)$ can be bounded by $T(1 - \gamma)sp(v^*)$.

**Bounding (b)** Using Lemma 3.2 that controls the difference between $\widetilde{V}_u^{t+1}$ and $\widetilde{V}_u^t$, we have

$$\begin{aligned} V_{t+1}^t(s_{t+1}) &= \text{CLIP}(\widetilde{V}_{t+1}^t(s_{t+1}); m_t, m_t + H) \\ &\leq \text{CLIP}(\widetilde{V}_{t+1}^t(s_{t+1}); m_{t+1}, m_{t+1} + H) + m_t - m_{t+1} \\ &\leq \widetilde{V}_{t+1}^t(s_{t+1}) + m_t - m_{t+1} \\ &\leq \widetilde{V}_{t+1}^{t+1}(s_{t+1}) + 2m_{t-1} - 2m_{t+1} \end{aligned}$$

where the second inequality holds because $\widetilde{V}_{t+1}^t(s_{t+1}) \geq m_{t+1}$ by Line 15. Hence, term $(b)$ can be bounded by $\mathcal{O}(\frac{1}{1-\gamma})$ using telescoping sums of $\widetilde{V}_{t+1}^{t+1}(s_{t+1}) - \widetilde{V}_t^t(s_t)$ and $2m_{t-1} - 2m_{t+1}$, and the fact that $V_u^t \leq \frac{1}{1-\gamma}$ and $m_t \leq \frac{1}{1-\gamma}$ for all $t \in [T]$ and $u \in [t : T]$.

**Bounding (c)** Since $V_u^t$ is $\mathcal{F}_t$-measurable where $\mathcal{F}_t$ is history up to time step $t$, we have $\mathbb{E}[V_{t+1}^t(s_{t+1})|\mathcal{F}_t] = [PV_{t+1}^t](s_t, a_t)$, making the summation $(c)$ a summation of a martingale difference sequence. Since $sp(V_{t+1}^t) \leq H$ for all $t \in [T]$, the summation can be bounded by $\widetilde{\mathcal{O}}(sp(v^*)\sqrt{T})$ using Azuma-Hoeffding inequality.

**Bounding (d)** The sum of the bonus terms can be bounded by $\widetilde{\mathcal{O}}(\beta\sqrt{dT})$ using a standard analysis from literature on linear MDP.

Combining the bounds, and choosing $H = 2 \cdot sp(v^*)$ and $\beta = \widetilde{\mathcal{O}}(sp(v^*)d)$ specified in Lemma 3.3, we get

$$\begin{aligned} R_T \leq \widetilde{\mathcal{O}}(T(1-\gamma)sp(v^*) + \tfrac{1}{1-\gamma} + sp(v^*)\sqrt{T} \\ + sp(v^*)\sqrt{d^3T}). \end{aligned}$$

Choosing $\gamma = 1 - \sqrt{1/T}$, we get $R_T \leq \widetilde{\mathcal{O}}(sp(v^*)\sqrt{d^3T})$, leading to our main result (see Appendix C for a more detailed analysis):

**Theorem 3.6.** *Under Assumptions A and B, running Algorithm 2 with inputs $\gamma = 1 - \sqrt{1/T}$, $\lambda = 1$, $H = 2 \cdot sp(v^*)$ and $\beta = 2c_\beta \cdot sp(v^*)d\sqrt{\log(dT/\delta)}$ guarantees with probability at least $1 - \delta$,*

$$R_T \leq \mathcal{O}(sp(v^*)\sqrt{d^3T\log(dT/\delta)\log T}).$$

*where $c_\beta$ is defined in Lemma 3.3.*

The regret bound for our algorithm $\gamma$-DC-LSCVI-UCB matches the regret bound of the previous algorithm $\gamma$-LSCVI-UCB.

**Remark 3.7.** Both algorithms $\gamma$-DC-LSCVI-UCB and $\gamma$-LSCVI-UCB require the knowledge of the time horizon $T$ to tune the discount factor $\gamma$ in order to achieve a $T$-step regret bound of $\widetilde{\mathcal{O}}(\sqrt{T})$. This limitation can be addressed using the standard doubling trick, which allows us to obtain a regret bound of $\widetilde{\mathcal{O}}(\sqrt{T})$ for *any* horizon $T$. The doubling trick is a standard technique in online learning to convert an algorithm with $\mathcal{O}(\sqrt{T})$ regret guarantee for a fixed known $T$ to an *anytime* algorithm that does not take $T$ as an input and guarantee $T$-step regret of $\mathcal{O}(\sqrt{T})$ for any $T$. The idea is to run the algorithm in phases, where each phase lasts twice as long as the previous one. At the beginning of each phase, the algorithm is restarted with parameters tuned for that phase length.

### 3.4. Computational Complexity

Our algorithm $\gamma$-LSCVI-UCB+ runs up to $T$ steps of value iteration every time step, resulting in $\mathcal{O}(T^2)$ value iteration steps. This can be seen by the nested loop structure of the algorithm, where the outer loop is indexed by $t$ for the time step and the inner loop is indexed by $u$ for the value iteration step. The computational bottleneck of the algorithm is computing $\widetilde{Q}_u^t(s,a)$ for all $a \in \mathcal{A}$ and all $s \in \{s_1, \ldots, s_{t-1}\}$, which involves computing the regression coefficient $\widehat{w}_t(V_{u+1}^t)$. Computing the regression coefficient takes $\mathcal{O}(T + d^2)$ operations.

In total, the computational complexity of our algorithm is $\mathcal{O}(T^3 d^2 A)$, which is polynomial in the problem parameters $T, d, A$ and is independent of the size of the state space. Although our algorithm enjoys a polynomial-time computational complexity, it is super linear in $T$, just as the the OLSVI.FH algorithm (Wei et al., 2021) and the previous work $\gamma$-LSCVI-UCB (Hong et al., 2025). We leave further improving the computational complexity to be linear in $T$ as future work.

## 4. Conclusion

We propose an algorithm for infinite-horizon average-reward RL with linear MDPs that achieves a regret bound of $\widetilde{\mathcal{O}}(\mathrm{sp}(v^*)\sqrt{d^3 T})$ and is computationally efficient. Our algorithm uses a combination of techniques such as approximation by discounted setting, value function clipping for constraining its span, and deviation-controlled value iteration. An interesting future directions include improving the regret bound by a factor of $\sqrt{d}$ using variance-aware regression method, extending the techniques to the general function approximation setting, and learning a stationary policy for memory efficiency.

## Impact Statement

This paper presents work whose goal is to advance the field of Machine Learning. There are many potential societal consequences of our work, none which we feel must be specifically highlighted here.

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

## A. Concentration Inequalities

**Lemma A.1** (Concentration of vector-valued self-normalized processes (Abbasi-Yadkori et al., 2011)). *Let $\{\varepsilon_t\}_{t=1}^{\infty}$ be a real-valued stochastic process with corresponding filtration $\{\mathcal{F}_t\}_{t=0}^{\infty}$. Let $\varepsilon_t | \mathcal{F}_{t-1}$ be zero-mean and $\sigma$-subgaussian. Let $\{\phi_t\}_{t=0}^{\infty}$ be an $\mathbb{R}^d$-valued stochastic process where $\phi_t \in \mathcal{F}_{t-1}$. Assume $\Lambda_0$ is a $d \times d$ positive definite matrix, and let $\Lambda_t = \Lambda_0 + \sum_{s=1}^{t} \phi_s \phi_s^T$. Then for any $\delta > 0$, with probability at least $1 - \delta$, we have for all $t \geq 0$ that*

$$\left\| \sum_{s=1}^{t} \phi_s \varepsilon_s \right\|_{\Lambda_t^{-1}}^2 \leq 2\sigma^2 \log \left( \frac{det(\Lambda_t)^{1/2} det(\Lambda_0)^{-1/2}}{\delta} \right).$$

**Lemma A.2.** *Let $\boldsymbol{w}$ be a ridge regression coefficient obtained by regressing $y \in [0, B]$ on $\boldsymbol{x} \in \mathbb{R}^d$ using the dataset $\{(\boldsymbol{x}_i, y_i)\}_{i=1}^{n}$ so that $\boldsymbol{w} = \Lambda^{-1} \sum_{i=1}^{n} \boldsymbol{x}_i y_i$ where $\Lambda = \sum_{i=1}^{n} \boldsymbol{x} \boldsymbol{x}^T + \lambda I$. Then,*

$$\|\boldsymbol{w}\|_2 \leq B\sqrt{dn/\lambda}.$$

**Lemma A.3.** *Let $V : \mathcal{S} \to [-B, B]$ be a bounded function. Then, $\boldsymbol{w}^*(V) = \int_{\mathcal{S}} V(s') d\boldsymbol{\mu}(s')$ which satisfies $[PV](s, a) = \langle \varphi(s, a), \boldsymbol{w}^*(V) \rangle$ for all $(s, a) \in \mathcal{S} \times \mathcal{A}$, satisfies*

$$\|\boldsymbol{w}^*(V)\|_2 \leq B\sqrt{d}.$$

*Proof.*

$$\|\boldsymbol{w}^*(V)\|_2 = \left\| \int_{\mathcal{S}} V(s') d\boldsymbol{\mu}(s') \right\|_2 \leq B \left\| \int_{\mathcal{S}} d\boldsymbol{\mu}(s') \right\|_2 \leq B\sqrt{d}$$

where the first inequality holds since $\boldsymbol{\mu}$ is a vector of positive measures and $V(s') \geq 0$. The last inequality is by the boundedness assumption (1) on $\boldsymbol{\mu}(\mathcal{S})$. $\qquad \square$

**Lemma A.4** (Adaptation of Lemma D.4 in Jin et al. (2020)). *Let $\{x_t\}_{t=1}^{\infty}$ be a stochastic process on state space $\mathcal{S}$ with corresponding filtration $\{\mathcal{F}_t\}_{t=0}^{\infty}$. Let $\{\phi_t\}_{t=0}^{\infty}$ be a $\mathbb{R}^d$-valued stochastic process where $\phi_t \in \mathcal{F}_{t-1}$, and $\|\phi_t\|_2 \leq 1$. Let $\Lambda_n = \lambda I + \sum_{t=1}^{n} \phi_t \phi_t^T$. Then for any $\delta > 0$ and any given function class $\mathcal{V}$, with probability at least $1 - \delta$, for all $n \geq 0$, and any $V \in \mathcal{V}$ satisfying $sp(V) \leq H$, we have*

$$\left\| \sum_{t=1}^{n} \phi_t (V(x_t) - \mathbb{E}[V(x_t)|\mathcal{F}_{t-1}]) \right\|_{\Lambda_n^{-1}}^2 \leq 4H^2 \left[ \frac{d}{2} \log \left( \frac{n + \lambda}{\lambda} \right) + \log \frac{\mathcal{N}_\varepsilon}{\delta} \right] + \frac{8n^2 \varepsilon^2}{\lambda}$$

*where $\mathcal{N}_\varepsilon$ is the $\varepsilon$-covering number of $\mathcal{V}$ with respect to the distance $dist(V, V') = \sup_x |V(x) - V'(x)|$.*

**Lemma A.5** (Adaptation of Lemma B.3 in Jin et al. (2020)). *Under the linear MDP setting in Theorem 3.6 for the $\gamma$-LSCVI-UCB algorithm with clipping oracle (Algorithm 1), let $c_\beta$ be the constant in the definition of $\beta = c_\beta H d \sqrt{\log(dT/\delta)}$. There exists an absolute constant $C$ that is independent of $c_\beta$ such that for any fixed $\delta \in (0, 1)$, the event $\mathcal{E}$ defined by*

$$\forall u \in [T], \ t \in [T] :$$
$$\left\| \sum_{\tau=1}^{t-1} \varphi(s_\tau, a_\tau) [V_u^t(s_{\tau+1}) - [PV_u^t](s_\tau, a_\tau)] \right\|_{\Lambda_t^{-1}} \leq C \cdot H d \sqrt{\log((c_\beta + 1)dT/\delta)}$$

*satisfies $P(\mathcal{E}) \geq 1 - \delta$.*

*Proof.* By Lemma A.2, we have $\|\boldsymbol{w}_t\|_2 \leq H\sqrt{dt/\lambda}$ for all $t = 1, \ldots, T$. Hence, by combining Lemma D.3 and Lemma A.4, for any $\varepsilon > 0$ and any fixed pair $(u, t) \in [T] \times [T]$, we have with probability at least $1 - \delta/T^2$ that

$$\left\| \sum_{\tau=1}^{t-1} \varphi(s_\tau, a_\tau) [V_u^t(s_{\tau+1}) - [PV_u^t](s_\tau, a_\tau)] \right\|_{\Lambda_t^{-1}}^2$$
$$\leq 4H^2 \left[ \frac{2}{d} \log \left( \frac{t + \lambda}{\lambda} \right) + d \log \left( 1 + \frac{4H\sqrt{dt}}{\varepsilon\sqrt{\lambda}} \right) + d^2 \log \left( 1 + \frac{8d^{1/2}\beta^2}{\varepsilon^2 \lambda} \right) + \log \left( \frac{T^2}{\delta} \right) \right] + \frac{8t^2 \varepsilon^2}{\lambda}.$$

Using a union bound over $(u, t) \in [T] \times [T]$ and choosing $\varepsilon = Hd/t$ and $\lambda = 1$, there exists an absolute constant $C > 0$ independent of $c_\beta$ such that, with probability at least $1 - \delta$,

$$\left\| \sum_{\tau=1}^{t-1} \varphi(s_\tau, a_\tau)[V_u^t(s_{\tau+1}) - [PV_u^t](s_\tau, a_\tau)] \right\|_{\Lambda_t^{-1}}^2 \le C^2 \cdot d^2 H^2 \log((c_\beta + 1)dT/\delta),$$

which concludes the proof. $\qquad\square$

### A.1. Proof of Lemma 3.3

*Proof of Lemma 3.3.* We prove under the event $\mathcal{E}$ defined in Lemma A.5. Recall the definition

$$[\widehat{P}_t V_u^t](s, a) = \langle \varphi(s, a), \widehat{w}_t(V_u^t - V_u^t(s_1)) \rangle + V_u^t(s_1)$$

where $\widehat{w}_t(V_u^t - V_u^t(s_1)) = \Lambda_t^{-1} \sum_{\tau=1}^{t-1} (V_u^t(s_{\tau+1}) - V_u^t(s_1)) \cdot \varphi(s_\tau, a_\tau)$. For convenience, we introduce the notation $\bar{V}_u^k(s) = V_u^k(s) - V_u^k(s_1)$ and $w_u^t = \widehat{w}_t(\bar{V}_u^t)$. With these notations, we have

$$[\widehat{P}_t V_u^t](s, a) = \langle \varphi(s, a), w_u^t \rangle + V_u^t(s_1), \quad w_u^t = \Lambda_t^{-1} \sum_{\tau=1}^{t-1} \varphi(s_\tau, a_\tau) \bar{V}_u^k(s_{\tau+1}).$$

We can decompose $\langle \varphi(s, a), w_u^t \rangle$ as

$$\langle \varphi(s, a), w_u^t \rangle = \underbrace{\langle \varphi(s, a), \Lambda_t^{-1} \sum_{\tau=1}^{t-1} \varphi(s_\tau, a_\tau)[P\bar{V}_u^t](s_\tau, a_\tau) \rangle}_{(a)} + \underbrace{\langle \varphi(s, a), \Lambda_t^{-1} \sum_{\tau=1}^{t-1} \varphi(s_\tau, a_\tau)(\bar{V}_u^t(s_{\tau+1}) - [P\bar{V}_u^t](s_\tau, a_\tau)) \rangle}_{(b)}.$$

Since $\bar{V}_u^t(s) \in [-H, H]$ for all $s \in \mathcal{S}$, it follows by Lemma A.3 that $\|w^*(\bar{V}_u^t)\|_2 \le H\sqrt{d}$. Hence, the first term $(a)$ in the display above can be bounded as

$$\langle \varphi(s, a), \Lambda_t^{-1} \sum_{\tau=1}^{t-1} \varphi(s_\tau, a_\tau)[P\bar{V}_u^t](s_\tau, a_\tau) \rangle = \langle \varphi(s, a), \Lambda_t^{-1} \sum_{\tau=1}^{t-1} \varphi(s_\tau, a_\tau)\varphi(s_\tau, a_\tau)^T w^*(\bar{V}_u^t) \rangle$$

$$= \langle \varphi(s, a), w^*(\bar{V}_u^t) \rangle - \lambda \langle \varphi(s, a), \Lambda_t^{-1} w^*(\bar{V}_u^t) \rangle$$

$$\le \langle \varphi(s, a), w^*(\bar{V}_u^t) \rangle + \lambda \|\varphi(s, a)\|_{\Lambda_t^{-1}} \|w^*(\bar{V}_u^t)\|_{\Lambda_t^{-1}}$$

$$\le \langle \varphi(s, a), w^*(\bar{V}_u^t) \rangle + H\sqrt{\lambda d} \|\varphi(s, a)\|_{\Lambda_t^{-1}}$$

where the first inequality is by Cauchy-Schwartz and the second inequality is by Lemma A.3. Under the event $\mathcal{E}$ defined in Lemma A.5, the second term $(b)$ can be bounded by

$$\langle \varphi(s, a), \Lambda_t^{-1} \sum_{\tau=1}^{t-1} \varphi(s_\tau, a_\tau)(\bar{V}_u^t(s_{\tau+1}) - [P\bar{V}_u^t](s_\tau, a_\tau))$$

$$\le \|\varphi(s, a)\|_{\Lambda_t^{-1}} \left\| \sum_{\tau=1}^{t-1} \varphi(s_\tau, a_\tau)(V_u^t(s_{\tau+1}) - [PV_u^t](s_\tau, a_\tau)) \right\|_{\Lambda_t^{-1}}$$

$$\le C \cdot Hd\sqrt{\log((c_\beta + 1)dT/\delta)} \cdot \|\varphi(s, a)\|_{\Lambda_t^{-1}}.$$

Combining the two bounds and rearranging, we get

$$\langle \phi, w_u^t - w^*(\bar{V}_u^t) \rangle \le C \cdot Hd\sqrt{(\log(c_\beta + 1)dT/\delta)} \cdot \|\phi\|_{\Lambda_t^{-1}}$$

for some absolute constant $C$ independent of $c_\beta$. Lower bound of $\langle \phi, w_u^t - w^*(\bar{V}_u^t) \rangle$ can be shown similarly, establishing

$$|\langle \phi, w_u^t - w^*(\bar{V}_u^t) \rangle| \le C \cdot Hd\sqrt{\log((c_\beta + 1)dT/\delta)} \cdot \|\phi\|_{\Lambda_t^{-1}}.$$

Hence,

$$
\begin{aligned}
|[\widehat{P}_t V_u^t](s,a) - [P V_u^t](s,a)| &= |\langle \boldsymbol{\varphi}(s,a), \widehat{\boldsymbol{w}}_t(V_u^t - V_u^t(s_1)) \rangle + V_u^t(s_1) - \langle \boldsymbol{\varphi}(s,a), \boldsymbol{w}^*(V_u^t) \rangle| \\
&= |\langle \boldsymbol{\varphi}(s,a), \boldsymbol{w}_u^t - \boldsymbol{w}^*(\bar{V}_u^t) \rangle| \\
&\leq C \cdot H d \sqrt{\log((c_\beta + 1)dT/\delta)} \cdot \|\boldsymbol{\phi}\|_{\Lambda_t^{-1}}
\end{aligned}
$$

where the last equality uses the fact that $\boldsymbol{w}^*(V) = \int_{\mathcal{S}} V(s') \boldsymbol{\mu}(s')$ is linear. It remains to show that there exists a choice of absolute constant $c_\beta$ such that

$$
C \sqrt{\log(c_\beta + 1) + \log(dT/\delta)} \leq c_\beta \sqrt{\log(dT/\delta)}.
$$

Noting that $\log(dT/\delta) \geq \log 2$, this can be done by choosing an absolute constant $c_\beta$ that satisfies $C\sqrt{\log 2 + \log(c_\beta + 1)} \leq c_\beta \sqrt{\log 2}$. $\qquad \square$

**Lemma A.6.** *The clipping operation* $\mathrm{CLIP}(x; L, U)$ *has the following properties:*

(i) $\mathrm{CLIP}(x; L, U) = \mathrm{CLIP}(x - c; L - c, U - c) + c.$

(ii) $\mathrm{CLIP}(x; L, U) \leq \mathrm{CLIP}(y; L, U)$ *if* $x \leq y.$

(iii) $\mathrm{CLIP}(x; L, U) \leq x$ *if and only if* $x \geq L.$

(iv) $\mathrm{CLIP}(x; L, U) \geq \mathrm{CLIP}(x; L', U')$ *if* $L \geq L'$ *and* $U \geq U'.$

*Proof.* The proofs are straight from the definition. $\qquad \square$

# B. Deviation-Controlled Value Iteration

## B.1. Positive Result for Tabular MDPs

In this section, we show that the scheme used in the algorithm $\gamma$-LSCVI-UCB+ for controlling the deviation between chains of value functions with different clipping thresholds is not necessary in the tabular setting.

To reuse the notations developed for the linear setting, we treat the tabular setting with the size of the state space $S$ and the size of the action space $A$ as the $SA$-dimensional linear MDP setting where each pair $(s,a) \in \mathcal{S} \times \mathcal{A}$ is mapped to a one-hot encoded vector $\boldsymbol{\varphi}(s,a) = \boldsymbol{e}_{(s,a)} \in \mathbb{R}^{SA}$ where the entry associated to $(s,a)$ is equal to 1 and all other entries 0. We show that under the tabular setting, Algorithm 3 that removes the step for clipping $Q_u^t$ from $\gamma$-LSCVI-UCB+ successfully control the deviation of a chain of value functions from its previous chain. Note that the algorithm uses the doubling-trick that updates the covariance matrix used for regression only when its determinant doubles. The trick is used to facilitate the analysis of the difference $Q_u^t(s,a) - Q_u^{t+1}(s,a)$ shown in the proof of the lemma below.

We use $\lambda = 0$ and treat $\Lambda_t^{-1}$ as the pseudoinverse of $\Lambda$, and set $\|\boldsymbol{\varphi}(s,a)\|_{\Lambda_t^{-1}} = \frac{1}{1-\gamma}$ when $\|\boldsymbol{\varphi}(s,a)\|_{\Lambda_t^{-1}} = 0$, that is, when the direction $\boldsymbol{\varphi}(s,a)$ is never explored. Then, as shown in the following lemma, the deviation between chains of value iterations is controlled even without the extra scheme used for the linear MDP setting.

**Lemma B.1.** *When running* $\gamma$-LSCVI-UCB+ *algorithm without deviation control under the tabular setting, for all* $t \in [T]$, $u \in [t : T]$, *we have*

$$
\begin{aligned}
|\widetilde{V}_u^{t+1}(s) - \widetilde{V}_u^t(s)| &\leq m_t - m_{t+1} \\
|V_u^{t+1}(s) - V_u^t(s)| &\leq m_t - m_{t+1}
\end{aligned}
$$

*for all* $s \in \mathcal{S}$.

*Proof.* We introduce the notation $N_t(s,a) = \sum_{\tau=1}^{t-1} \mathbb{I}\{s_\tau = s, a_\tau = a\}$ and $N_t(s,a,s') = \sum_{\tau=1}^{t-1} \mathbb{I}\{s_\tau = s, a_\tau = a, s_{\tau+1} = s'\}$, which is the visitation counts up to (excluding) time step $t$ of the state-action pair $(s,a)$ and state-action-state triplet $(s,a,s')$, respectively. Note that in the tabular setting, we have $[\widehat{P}_t V](s,a) =$

---

**Algorithm 3** $\gamma$-LSCVI-UCB+ without Deviation Control

---

**Input:** Discounting factor $\gamma \in [0, 1)$, regularization constant $\lambda > 0$, span $H > 0$, bonus factor $\beta > 0$.

**Initialize:** $k \leftarrow 1, t_k \leftarrow 1, \Lambda_1 \leftarrow \lambda I, m_1 \leftarrow \frac{1}{1-\gamma}$.

1: Receive state $s_1$.
2: **for** $t = 1, \ldots, T$ **do**
3:     $V_{T+1}^t(\cdot) \leftarrow \frac{1}{1-\gamma}$.
4:     **for** $u = T, T-1, \ldots, t$ **do**
5:        $Q_u^t(\cdot, \cdot) \leftarrow \left( r(\cdot, \cdot) + \gamma([\widehat{P}_{t_k} V_{u+1}^t](\cdot, \cdot) + \beta \|\varphi(\cdot, \cdot)\|_{\Lambda_{t_k}^{-1}}) \right) \wedge \frac{1}{1-\gamma}$.
6:        $\widetilde{V}_u^t(\cdot) \leftarrow \max_a Q_u^t(\cdot, a)$.
7:        $V_u^t(\cdot) \leftarrow \text{CLIP}(\widetilde{V}_u^t(\cdot); m_t, m_t + H)$.
8:     **end for**
9:     Take action $a_t \leftarrow \text{argmax}_{a \in \mathcal{A}} Q_t^t(s_t, a)$. Receive reward $r(s_t, a_t)$. Receive next state $s_{t+1}$.
10:    $\Lambda_{t+1} \leftarrow \Lambda_t + \varphi(s_t, a_t)\varphi(s_t, a_t)^\top$.
11:    $m_{t+1} \leftarrow \widetilde{V}_{t+1}^t(s_{t+1}) \wedge m_t$.
12:    **if** $2\det(\Lambda_{t_k}) < \det(\Lambda_{t+1})$ **then**
13:      $k \leftarrow k + 1, t_k \leftarrow t + 1$.
14:    **end if**
15: **end for**

---

$\sum_{s':N_t(s,a,s')>0}(N_t(s,a,s')/N_t(s,a))V(s')$, which is the expectation of $V$ with respect to the empirical transition probability kernel $\widehat{P}_t$: $\widehat{P}_t(s'|s,a) = N_t(s,a,s')/N_t(s,a)$. Hence, $\widehat{P}_t$ is linear such that $[\widehat{P}_t V_1](s,a) - [\widehat{P}_t V_2](s,a) = [\widehat{P}_t(V_1 - V_2)](s,a)$, and it satisfies $[\widehat{P}_t \Delta](s,a) \leq \|\Delta\|_\infty$ for any function $\Delta : \mathcal{S} \to \mathbb{R}$. We exploit these facts to prove the lemma.

We show by induction on $u = T+1, \ldots, 1$. Fix $t$ such that both $t$ and $t+1$ are in the same episode $k$. For the base case $u = T+1$, we have $V_{T+1}^{t+1}(s) = V_{T+1}^t(s) = \frac{1}{1-\gamma}$ for all $s \in \mathcal{S}$, and trivially, we have $|V_{T+1}^{t+1}(s) - V_{T+1}^t(s)| \leq m_t - m_{t+1}$. Now, suppose $|V_{u+1}^{t+1}(s) - V_{u+1}^t(s)| \leq m_t - m_{t+1}$ for all $s \in \mathcal{S}$ for some $u \in [T]$. Then,

$$|Q_u^t(s,a) - Q_u^{t+1}(s,a)| \leq \gamma([\widehat{P}_{t_k} V_{u+1}^t](s,a) - [\widehat{P}_{t_k} V_{u+1}^{t+1}](s,a)) \leq m_t - m_{t+1}$$

where the first inequality is by the fact that $(\cdot \wedge \frac{1}{1-\gamma})$ is a contraction and the second inequality is by the previous discussion on $\widehat{P}_{t_k}$ being a expectation with respect to a proper probability kernel in the tabular setting. Since $\max_a Q(\cdot, a)$ is a contraction, it follows that $|\widetilde{V}_u^t(s) - \widetilde{V}_u^{t+1}(s)| \leq m_t - m_{t+1}$. Hence, using the fact that $m_t \geq m_{t+1}$, we have

$$
\begin{aligned}
V_u^t(s) - V_u^{t+1}(s) &= \text{CLIP}(\widetilde{V}_u^t(s); m_t, m_t + H) - \text{CLIP}(\widetilde{V}_u^{t+1}(s); m_{t+1}, m_{t+1} + H) \\
&\leq \text{CLIP}(\widetilde{V}_u^{t+1}(s) + m_t - m_{t+1}; m_t, m_t + H) - \text{CLIP}(\widetilde{V}_u^{t+1}(s); m_{t+1}, m_{t+1} + H) \\
&= \text{CLIP}(\widetilde{V}_u^{t+1}(s); m_{t+1}, m_{t+1} + H) + m_t - m_{t+1} - \text{CLIP}(\widetilde{V}_u^{t+1}(s); m_{t+1}, m_{t+1} + H) \\
&= m_t - m_{t+1}
\end{aligned}
$$

where the second equality uses the property (i) of the clipping operation. Similarly, we have

$$
\begin{aligned}
V_u^t(s) - V_u^{t+1}(s) &= \text{CLIP}(\widetilde{V}_u^t(s); m_t, m_t + H) - \text{CLIP}(\widetilde{V}_u^{t+1}(s); m_{t+1}, m_{t+1} + H) \\
&\geq \text{CLIP}(\widetilde{V}_u^t(s); m_t, m_t + H) - \text{CLIP}(\widetilde{V}_u^{t+1}(s); m_t, m_t + H) \\
&\geq \text{CLIP}(\widetilde{V}_u^t(s); m_t, m_t + H) - \text{CLIP}(\widetilde{V}_u^t(s) - m_t + m_{t+1}; m_t, m_t + H) \\
&= \text{CLIP}(\widetilde{V}_u^t(s); m_t, m_t + H) - \text{CLIP}(\widetilde{V}_u^t(s); m_{t+1}, m_{t+1} + H) - m_t + m_{t+1} \\
&\geq -m_t + m_{t+1}
\end{aligned}
$$

where the second equality uses the property (i) of the clipping operation. The two inequalities establish $|V_u^t(s) - V_u^{t+1}(s)| \leq m_t - m_{t+1}$ as desired. By induction, the proof is complete. $\square$

### B.2. Negative Result for Linear MDPs

*Proof of Lemma 3.1.* For convenience, let $n = 2m$. If $n$ is odd, we can take $\phi_n = \mathbf{0}$ and similar argument holds. Take $\phi_1, \dots \phi_m = (\eta, 1/2, 0, \dots, 0)$ and $\phi_{m+1}, \dots, \phi_{2m} = (\eta, -1/2, 0, \dots, 0)$ where $\eta > 0$ is to be chosen later. Take $y_1 = \cdots = y_{2m} = \Delta$ and $\lambda = 1$. Then, $\Lambda_n = \text{diag}(\eta^2 n, n/4, 0, \dots, 0) + I$ and $\sum_{i=1}^n y_i \phi_i = (\eta \Delta n, 0, \dots, 0)$. Hence, $\boldsymbol{w}_n = (\frac{\eta \Delta n}{\eta^2 n+1}, 0, \dots, 0)$. It follows that, choosing $\phi = (1, 0, \dots, 0)$, we get

$$|\langle \boldsymbol{w}_n, \phi \rangle| = \frac{\eta \Delta n}{\eta^2 n + 1}.$$

Choosing $\eta = 1/\sqrt{n}$, we get $|\langle \boldsymbol{w}_n, \phi \rangle| = \frac{1}{2}\Delta\sqrt{n}$, which completes the proof. □

### B.3. Deviation-Controlled Value Iteration for Linear MDPs

**Lemma B.2.** *For all $t \in [T]$, $u \in [t : T]$, we have*

$$|\widetilde{V}_u^{t+1}(s) - \widetilde{V}_u^t(s)| \le m_{t-1} - m_{t+1}$$
$$|V_u^{t+1}(s) - V_u^t(s)| \le m_{t-1} - m_{t+1}$$

*for all $s \in \mathcal{S}$.*

*Proof.* We first show that $\widetilde{V}_u^{t+1}(s) - \widetilde{V}_u^t(s) \ge -m_{t-1} + m_{t+1}$ and $V_u^{t+1}(s) - V_u^t(s) \ge -m_{t-1} + m_{t+1}$. By definitions of $Q_u^{t+1}$ and $Q_u^t$, we have

$$
\begin{aligned}
Q_u^{t+1}(s, a) &= \text{CLIP}(\widetilde{Q}_u^{t+1}(s, a); L_u^{t+1}(s, a), U_u^{t+1}(s, a)) \\
&\ge L_u^{t+1}(s, a) \\
&= (\widetilde{Q}_u^t(s, a) - m_t + m_{t+1}) \vee (\widetilde{Q}_u^{t-1}(s, a) - m_{t-1} + m_{t+1}) \\
&\ge \widetilde{Q}_u^{t-1}(s, a) - m_{t-1} + m_{t+1},
\end{aligned}
$$

and

$$
\begin{aligned}
Q_u^t(s, a) &= \text{CLIP}(\widetilde{Q}_u^t(s, a); L_u^t(s, a), U_u^t(s, a)) \\
&\le U_u^t(s, a) \\
&= \widetilde{Q}_u^{t-1}(s, a) \wedge \widetilde{Q}_u^{t-2}(s, a) \\
&\le \widetilde{Q}_u^{t-1}(s, a).
\end{aligned}
$$

Chaining the two inequalities, we get $Q_u^{t+1}(s, a) \ge Q_u^t(s, a) - m_{t-1} + m_{t+1}$. It follows that

$$
\begin{aligned}
\widetilde{V}_u^{t+1}(s) &= \max_a Q_u^{t+1}(s, a) \\
&\ge \max_a Q_u^t(s, a) - m_{t-1} + m_{t+1} \\
&= \widetilde{V}_u^t(s) - m_{t-1} + m_{t+1},
\end{aligned}
$$

which shows the first claim. Hence,

$$
\begin{aligned}
V_u^{t+1}(s) &= \text{CLIP}(\widetilde{V}_u^{t+1}(s); m_{t+1}, m_{t+1} + H) \\
&\ge \text{CLIP}(\widetilde{V}_u^t(s) - m_{t-1} + m_{t+1}; m_{t+1}, m_{t+1} + H) \\
&= \text{CLIP}(\widetilde{V}_u^t(s); m_{t-1}, m_{t-1} + H) - m_{t-1} + m_{t+1} \\
&\ge \text{CLIP}(\widetilde{V}_u^t(s); m_t, m_t + H) - m_{t-1} + m_{t+1} \\
&= V_u^t(s) - m_{t-1} + m_{t+1},
\end{aligned}
$$

where the second equality is by Property (i) of the clipping operation and the second inequality is by Property (iv) of the clipping operation and the fact that $m_{t-1} \ge m_t$. This shows the second claim.

Now, we show that $\widetilde{V}_u^{t+1}(s) - \widetilde{V}_u^t(s) \le m_{t-1} - m_{t+1}$ and $V_u^{t+1}(s) - V_u^t(s) \le m_{t-1} - m_{t+1}$. By definitions of $Q_u^{t+1}$ and $Q_u^t$, we have

$$
\begin{aligned}
Q_u^{t+1}(s,a) &= \text{CLIP}(\widetilde{Q}_u^{t+1}(s,a); L_u^{t+1}(s,a), U_u^{t+1}(s,a)) \\
&\le U_u^{t+1}(s,a) \\
&= \widetilde{Q}_u^t(s,a) \wedge \widetilde{Q}_u^{t-1}(s,a) \\
&\le \widetilde{Q}_u^{t-1}(s,a),
\end{aligned}
$$

and

$$
\begin{aligned}
Q_u^t(s,a) &= \text{CLIP}(\widetilde{Q}_u^t(s,a); L_u^t(s,a), U_u^t(s,a)) \\
&\ge L_u^t(s,a) \\
&= (\widetilde{Q}_u^{t-1}(s,a) - m_t + m_{t+1}) \vee (\widetilde{Q}_u^{t-2}(s,a) - m_{t-1} + m_{t+1}) \\
&\ge \widetilde{Q}_u^{t-1}(s,a) - m_t + m_{t+1} \\
&\ge \widetilde{Q}_u^{t-1}(s,a) - m_{t-1} + m_{t+1}.
\end{aligned}
$$

Chaining the two inequalities, we get $Q_u^{t+1}(s,a) \le Q_u^t(s,a) + m_{t-1} - m_{t+1}$, and it follows that

$$
\begin{aligned}
\widetilde{V}_u^{t+1} &= \max_a Q_u^{t+1}(s,a) \\
&\le \max_a Q_u^t(s,a) + m_{t-1} - m_{t+1} \\
&= \widetilde{V}_u^t(s) + m_{t-1} - m_{t+1},
\end{aligned}
$$

which shows the first claim. Hence,

$$
\begin{aligned}
V_u^{t+1}(s) &= \text{CLIP}(\widetilde{V}_u^{t+1}(s); m_{t+1}, m_{t+1} + H) \\
&\le \text{CLIP}(\widetilde{V}_u^t(s) + m_t - m_{t+1}; m_{t+1}, m_{t+1} + H) \\
&\le \text{CLIP}(\widetilde{V}_u^t(s) + m_t - m_{t+1}; m_t, m_t + H) \\
&= \text{CLIP}(\widetilde{V}_u^t(s); m_{t+1}, m_{t+1} + H) + m_t - m_{t+1} \\
&\le \text{CLIP}(\widetilde{V}_u^t(s); m_t, m_t + H) + m_t - m_{t+1} \\
&= V_u^t(s) + m_t - m_{t+1} \\
&\le V_u^t(s) + m_{t-1} - m_{t+1}.
\end{aligned}
$$

$\square$

## C. Regret Analysis

We first prove the optimism result that says the value function estimates are optimistic estimates of the true value function.

### C.1. Proof of Lemma 3.4

*Proof of Lemma 3.4.* We prove under the event $\mathcal{E}$ defined in Lemma A.5, which holds with probability at least $1 - \delta$. We prove by induction on $t$ and $u$.

Suppose $V_u^\tau(s) \ge V^*(s)$, $\widetilde{V}_u^\tau(s) \ge V^*(s)$ and $\widetilde{Q}_u^\tau(s,a) \ge Q^*(s,a)$ hold for all $\tau = 1, \ldots, t-1$ and $u \in [\tau : T]$ and $(s,a) \in \mathcal{S} \times \mathcal{A}$. If we show that $V_u^t(s) \ge V^*(s)$, $\widetilde{V}_u^t(s)$ and $\widetilde{Q}_u^t(s,a) \ge Q^*(s,a)$ for all $u \in [t : T]$ and $(s,a) \in \mathcal{S} \times \mathcal{A}$, the proof is complete by induction on $t$. We show this by induction on $u = T + 1, T, \ldots, t$.

The base case $u = T + 1$ holds since $V_{T+1}^t(s) = \frac{1}{1-\gamma} \ge V^*(s)$ for all $s \in \mathcal{S}$. Now, suppose $V_{u+1}^t(s) \ge V^*(s)$ for all

$s \in \mathcal{S}$ for some $u \in [t+1 : T]$. Then,

$$\widetilde{Q}_u^t(s,a) = (r(s,a) + \gamma([\widehat{P}_t V_{u+1}^t](s,a) + \beta\|\varphi(s,a)\|_{\Lambda_t^{-1}}) \wedge \frac{1}{1-\gamma}$$

$$\geq (r(s,a) + \gamma[PV_{u+1}^t](s,a)) \wedge \frac{1}{1-\gamma}$$

$$\geq (r(s,a) + \gamma[PV^*](s,a)) \wedge \frac{1}{1-\gamma}$$

$$= Q^*(s,a) \wedge \frac{1}{1-\gamma}$$

$$= Q^*(s,a)$$

where the first inequality is by the event $\mathcal{E}$, the second inequality by the induction hypothesis. The second equality is by the Bellman optimality equation. This shows $\widetilde{Q}_u^t(s,a) \geq Q^*(s,a)$ for all $(s,a) \in \mathcal{S} \times \mathcal{A}$ as desired. Additionally,

$$Q_u^t(s,a) = \text{CLIP}(\widetilde{Q}_u^t(s,a); L_u^t(s,a), U_u^t(s,a))$$

$$\geq \text{CLIP}(Q^*(s,a); L_u^t(s,a), U_u^t(s,a))$$

$$\geq Q^*(s,a) \wedge U_u^t(s,a)$$

$$= Q^*(s,a) \wedge (\widetilde{Q}_u^{t-1}(s,a) \wedge \widetilde{Q}_u^{t-2}(s,a))$$

$$\geq Q^*(s,a)$$

where the second inequality is by the clipping property (ii), and the last inequality holds by induction hypothesis. It follows that

$$\widetilde{V}_u^t(s) = \max_a Q_u^t(s,a) \geq \max_a Q^*(s,a) = V^*(s).$$

Note that by induction hypothesis, $\widetilde{V}_u^\tau(s) \geq V^*(s)$ for all $\tau \in [t-1]$, $u \in [\tau : T]$ and $s \in \mathcal{S}$. Hence, $m_t = \min\{\widetilde{V}_t^{t-1}(s_t), \widetilde{V}_{t-1}^{t-2}(s_{t-1}), \ldots, \widetilde{V}_2^1(s_2) \geq \min\{V^*(s_t), V^*(s_{t-1}), \ldots, V^*(s_2), \frac{1}{1-\gamma}\} \geq \min_{s \in \mathcal{S}} V^*(s)$. It follows that

$$V_u^t(s) = \text{CLIP}(\widetilde{V}_u^t(s); m_t, m_t + H)$$

$$\geq \text{CLIP}(V^*(s); m_t, m_t + H)$$

$$\geq \text{CLIP}(V^*(s); \min_{s' \in \mathcal{S}} V^*(s'), \min_{s' \in \mathcal{S}} V^*(s') + H)$$

$$\geq V^*(s)$$

where the last inequality uses the fact that $H \geq 2 \cdot \text{sp}(v^*)$ is chosen such that $\text{sp}(V^*) \leq H$. We have shown that if $V_{u+1}^t(s) \geq V^*(s)$ holds for all $s \in \mathcal{S}$, then $V_u^t(s) \geq V^*(s)$, $\widetilde{V}_u^t(s)$ and $\widetilde{Q}_u^t(s,a)$ hold for all $(s,a) \in \mathcal{S} \times \mathcal{A}$. By induction on $u = T, \ldots, 1$, it follows that $V_u^t(s) \geq V^*(s)$, $\widetilde{V}_u^t(s) \geq V^*(s)$ and $\widetilde{Q}_u^t(s,a) \geq Q^*(s,a)$ hold for all $(s,a) \in \mathcal{S} \times \mathcal{A}$. The proof is complete by induction on $t$. $\square$

Now, we show an upper bound of the action value function estimate, which is a direct consequence of the concentration inequality in Lemma A.5.

### C.2. Proof of Lemma 3.5

*Proof of Lemma 3.5.* We prove under the event $\mathcal{E}$ defined in Lemma A.5, which holds with probability at least $1 - \delta$. Fix any $t \in [T]$ and $u \in [t : T]$. By event $\mathcal{E}$, we have

$$\widetilde{Q}_u^t(s,a) = \left(r(s,a) + \gamma([\widehat{P}_t V_{u+1}^t](s,a) + \beta\|\varphi(\cdot,\cdot)\|_{\Lambda_t^{-1}}\right) \wedge \frac{1}{1-\gamma}$$

$$\leq r(s,a) + \gamma[PV_{u+1}^t](s,a) + 2\beta\|\varphi(s,a)\|_{\Lambda_t^{-1}}$$

for all $t \in [T]$. Hence, by Lemma B.2, we have for $t \geq 4$ that

$$
\begin{aligned}
\widetilde{Q}_u^{t-2}(s,a) &\leq r(s,a) + \gamma[PV_{u+1}^{t-2}](s,a) + 2\beta\|\varphi(s,a)\|_{\Lambda_t^{-1}} \\
&\leq r(s,a) + \gamma[P(V_{u+1}^t)](s,a) + 2\beta\|\varphi(s,a)\|_{\Lambda_t^{-1}} + m_{t-3} - m_{t-1} + m_{t-2} - m_t \\
&\leq r(s,a) + \gamma[PV_{u+1}^t](s,a) + 2\beta\|\varphi(s,a)\|_{\Lambda_t^{-1}} + 2(m_{t-3} - m_t).
\end{aligned}
$$

Therefore, for $t \geq 4$, we have

$$
\begin{aligned}
Q_u^t(s,a) &= \text{CLIP}(\widetilde{Q}_u^t(s,a); L_u^t(s,a), U_u^t(s,a)) \\
&\leq U_u^t(s,a) \\
&= \widetilde{Q}_u^{t-1}(s,a) \wedge \widetilde{Q}_u^{t-2}(s,a) \\
&\leq r(s,a) + \gamma[PV_{u+1}^t](s,a) + 2\beta\|\varphi(s,a)\|_{\Lambda_t^{-1}} + 2(m_{t-3} - m_t)
\end{aligned}
$$

$\square$

Finally, the following lemma will be used for bounding the sum of the bonus terms.

**Lemma C.1** (Lemma 11 in Abbasi-Yadkori et al. (2011)). *Let $\{\phi_t\}_{t \geq 1}$ be a bounded sequence in $\mathbb{R}^d$ with $\|\phi_t\|_2 \leq 1$ for all $t \geq 1$. Let $\Lambda_0 = I$ and $\Lambda_t = \sum_{i=1}^t \phi_i \phi_i^T + I$ for $t \geq 1$. Then,*

$$
\sum_{i=1}^t \phi_i^T \Lambda_{i-1}^{-1} \phi_i \leq 2\log\det(\Lambda_t) \leq 2d\log(1+t).
$$

### C.3. Proof of Main Theorem

Now, we are ready to prove the main theorem.

*Proof of Theorem 3.6.* We prove under the event $\mathcal{E}$ defined in Lemma A.5, which occurs with probability at least $1 - \delta$. By Lemma 3.5, we have for $t \geq 4$,

$$
Q_u^t(s,a) \leq r(s,a) + \gamma[PV_{u+1}^t](s,a) + 2\beta\|\varphi(s,a)\|_{\Lambda_t^{-1}} + 2(m_{t-3} - m_t).
$$

Plugging in $u \leftarrow t$, $s \leftarrow s_t$, $a \leftarrow a_t$, we get

$$
\begin{aligned}
R_T &= \sum_{t=1}^T (J^* - r(s_t, a_t)) \\
&\leq \sum_{t=4}^T (J^* - Q_t^t(s_t, a_t) + \gamma[PV_{t+1}^t](s_t, a_t) + 2\beta\|\varphi(s_t, a_t)\|_{\Lambda_t^{-1}} + 2(m_{t-3} - m_t)) + \mathcal{O}(1) \\
&= \underbrace{\sum_{t=4}^T (J^* - (1-\gamma)V_{t+1}^t(s_{t+1}))}_{(a)} + \underbrace{\sum_{t=4}^T (V_{t+1}^t(s_{t+1}) - Q_t^t(s_t, a_t))}_{(b)} \\
&\quad + \gamma \underbrace{\sum_{t=4}^T ([PV_{t+1}^t](s_t, a_t) - V_{t+1}^t(s_{t+1}))}_{(c)} + 2\beta \underbrace{\sum_{t=4}^T \|\varphi(s_t, a_t)\|_{\Lambda_t^{-1}}}_{(d)} + \mathcal{O}\left(\frac{1}{1-\gamma}\right).
\end{aligned}
$$

**Bounding (a)** By the optimism result (Lemma 3.4), we have $V_u^t(s) \geq V^*(s)$ for all $t \in [T]$ and $u \in [t:T]$ with high probability. It follows that

$$
\begin{aligned}
J^* - (1-\gamma)V_{t+1}^t(s_{t+1}) &\leq J^* - (1-\gamma)V^*(s_{t+1}) \\
&\leq (1-\gamma)\text{sp}(v^*)
\end{aligned}
$$

where the last inequality is by the bound on the error of approximating the average-reward setting by the discounted setting provided in Lemma 2.3. Hence, the term $(a)$ can be bounded by $T(1-\gamma)\text{sp}(v^*)$.

**Bounding (b)** Using Lemma 3.2 that controls the difference between $\widetilde{V}_u^{t+1}$ and $\widetilde{V}_u^t$, we have

$$
\begin{aligned}
V_{t+1}^t(s_{t+1}) &= \mathrm{CLIP}(\widetilde{V}_{t+1}^t(s_{t+1}); m_t, m_t + H) \\
&= \mathrm{CLIP}(\widetilde{V}_{t+1}^t(s_{t+1}) - m_t + m_{t+1}; m_{t+1}, m_{t+1} + H) + m_t - m_{t+1} \\
&\leq \mathrm{CLIP}(\widetilde{V}_{t+1}^t(s_{t+1}); m_{t+1}, m_{t+1} + H) + m_t - m_{t+1} \\
&\leq \widetilde{V}_{t+1}^t(s_{t+1}) + m_t - m_{t+1} \\
&\leq \widetilde{V}_{t+1}^{t+1}(s_{t+1}) + 2m_{t-1} - 2m_{t+1}
\end{aligned}
$$

where the second inequality holds because $\widetilde{V}_{t+1}^t(s_{t+1}) \geq m_{t+1}$ by Line 15. Hence, Term $(b)$ can be bounded by $\mathcal{O}(\frac{1}{1-\gamma})$ using telescoping sums of $\widetilde{V}_{t+1}^{t+1}(s_{t+1}) - \widetilde{V}_t^t(s_t)$ and $2m_{t-1} - 2m_{t+1}$, and the fact that $V_u^t \leq \frac{1}{1-\gamma}$ and $m_t \leq \frac{1}{1-\gamma}$ for all $t \in [T]$ and $u \in [t:T]$.

**Bounding (c)** Since $V_u^t$ is $\mathcal{F}_t$-measurable where $\mathcal{F}_t$ is history up to time step $t$, we have $\mathbb{E}[V_{t+1}^t(s_{t+1})|\mathcal{F}_t] = [PV_{t+1}^t](s_t, a_t)$, making the summation $(c)$ a summation of a martingale difference sequence. Since $\mathrm{sp}(V_{t+1}^t) \leq H$ for all $t \in [T]$, the summation can be bounded by $\mathcal{O}(\mathrm{sp}(v^*)\sqrt{T\log(1/\delta)})$ using Azuma-Hoeffding inequality.

**Bounding (d)** The sum of the bonus terms can be bounded by

$$
\begin{aligned}
\beta \sum_{t=1}^T \|\varphi(s_t, a_t)\|_{\Lambda_t^{-1}} &\leq \beta\sqrt{T}\left(\sum_{t=1}^T \|\varphi(s_t, a_t)\|_{\Lambda_t^{-1}}^2\right)^{1/2} \\
&\leq \mathcal{O}(\beta\sqrt{dT\log T})
\end{aligned}
$$

where the first inequality is by Cauchy-Schwartz and the last inequality is by Lemma C.1.

Combining the four bounds, and choosing $H = 2 \cdot \mathrm{sp}(v^*)$ and choosing $\beta = \mathcal{O}(\mathrm{sp}(v^*)d\sqrt{\log(dT/\delta)})$ specified in Lemma 3.3, we get

$$
R_T \leq \mathcal{O}(T(1-\gamma)\mathrm{sp}(v^*) + \tfrac{1}{1-\gamma} + \mathrm{sp}(v^*)\sqrt{T\log(1/\delta)} + \mathrm{sp}(v^*)\sqrt{d^3 T\log(dT/\delta)\log T}).
$$

Choosing $\gamma$ such that $\frac{1}{1-\gamma} = \sqrt{T}$, we get

$$
R_T \leq \mathcal{O}(\mathrm{sp}(v^*)\sqrt{d^3 T\log(dT/\delta)\log T}).
$$

$\square$

# D. Covering Numbers

In this section, we provide results on covering numbers of function classes used in this paper. We use the notation $\mathcal{N}_\epsilon(\mathcal{F}, \|\cdot\|)$ to denote the $\varepsilon$-covering number of the function class $\mathcal{F}$ with respect to the distance measure induced by the norm $\|\cdot\|$.

We first present a classical result that bounds the covering number of Euclidean ball.

**Lemma D.1.** *For any $\varepsilon > 0$, the $d$-dimensional Euclidean ball $\mathbb{B}_d(R)$ with radius $R > 0$ has log-covering number upper bounded by*

$$
\log \mathcal{N}_\varepsilon(\mathbb{B}_d(R), \|\cdot\|_2) \leq d\log(1 + 2R/\varepsilon).
$$

Using this classical result, we bound the covering number of the function class that captures the functions $\widetilde{Q}_u^t(\cdot, \cdot)$ encountered by our algorithm.

**Lemma D.2** (Adaptation of Lemma D.6 in Jin et al. (2020)). *Let $\mathcal{Q}$ be a class of functions mapping from $\mathcal{S} \times \mathcal{A}$ to $\mathbb{R}$ with the following parametric form*

$$
Q(\cdot, \cdot) = (\boldsymbol{w}^T \boldsymbol{\varphi}(\cdot, \cdot) + v + \beta\sqrt{\boldsymbol{\varphi}(\cdot, \cdot)^T \Lambda^{-1} \boldsymbol{\varphi}(\cdot, \cdot)}) \wedge M \tag{2}
$$

*where the parameters $(\boldsymbol{w}, \beta, v, \Lambda)$ satisfy $\|\boldsymbol{w}\| \leq L$, $\beta \in [0, B]$ and $v \in [0, D]$, and $\Lambda$ is a positive definite matrix with minimum eigenvalue satisfying $\lambda_{min}(\Lambda) \geq \lambda > 0$. The constant $M > 0$ is fixed. Assume $\|\boldsymbol{\varphi}(s, a)\| \leq 1$ for all $(s, a)$ pairs. Then*

$$\log \mathcal{N}_\varepsilon(\mathcal{Q}, \|\cdot\|_\infty) \leq d \log(1 + 8L/\varepsilon) + \log(1 + 8D/\varepsilon) + d^2 \log[1 + 8d^{1/2}B^2/(\lambda\varepsilon^2)].$$

*Proof.* Introducing $\boldsymbol{A} = \beta^2 \Lambda^{-1}$, we can reparameterize as

$$Q(\cdot, \cdot) = (\boldsymbol{w}^T \boldsymbol{\varphi}(\cdot, \cdot) + v + \sqrt{\boldsymbol{\varphi}(\cdot, \cdot)^T \boldsymbol{A} \boldsymbol{\varphi}(\cdot, \cdot)}) \wedge M$$

where the parameters $(\boldsymbol{w}, v, \boldsymbol{A})$ satisfy $\|\boldsymbol{w}\|_2 \leq L$, $\|\boldsymbol{A}\| \leq B^2\lambda^{-1}$, $v \in [0, D]$. For any pair of functions $Q_1, Q_2 \in \mathcal{Q}$ with parameterization $(\boldsymbol{w}_1, v_1, \boldsymbol{A}_1)$ and $(\boldsymbol{w}_2, v_2, \boldsymbol{A}_2)$, respectively, using the fact that $\cdot \wedge M$ is a contraction, we get

$$
\begin{aligned}
\|Q_1 - Q_2\|_\infty &\leq \sup_{s,a} |(\boldsymbol{w}_1^\top \boldsymbol{\varphi}(s, a) + v_1 + \sqrt{\boldsymbol{\varphi}(s, a)^\top \boldsymbol{A}_1 \boldsymbol{\varphi}(s, a)}) - (\boldsymbol{w}_2^\top \boldsymbol{\varphi}(s, a) + v_2 + \sqrt{\boldsymbol{\varphi}(s, a)^\top \boldsymbol{A}_2 \boldsymbol{\varphi}(s, a)})| \\
&\leq \sup_{\boldsymbol{\phi}: \|\boldsymbol{\phi}\|_2 \leq 1} |(\boldsymbol{w}_1^\top \boldsymbol{\phi} + v_1 + \sqrt{\boldsymbol{\phi}^\top \boldsymbol{A}_1 \boldsymbol{\phi}}) - (\boldsymbol{w}_2^\top \boldsymbol{\phi} + v_2 + \sqrt{\boldsymbol{\phi}^\top \boldsymbol{A}_2 \boldsymbol{\phi}})| \\
&\leq \sup_{\boldsymbol{\phi}: \|\boldsymbol{\phi}\|_2 \leq 1} |(\boldsymbol{w}_1 - \boldsymbol{w}_2)^\top \boldsymbol{\phi}| + |v_1 - v_2| + \sup_{\boldsymbol{\phi}: \|\boldsymbol{\phi}\|_2 \leq 1} \sqrt{|\boldsymbol{\phi}^\top (\boldsymbol{A}_1 - \boldsymbol{A}_2) \boldsymbol{\phi}|} \\
&= \|\boldsymbol{w}_1 - \boldsymbol{w}_2\|_2 + |v_1 - v_2| + \sqrt{\|\boldsymbol{A}_1 - \boldsymbol{A}_2\|_2} \\
&\leq \|\boldsymbol{w}_1 - \boldsymbol{w}_2\|_2 + |v_1 - v_2| + \sqrt{\|\boldsymbol{A}_1 - \boldsymbol{A}_2\|_F} \quad\quad\quad (3)
\end{aligned}
$$

where the third inequality uses the fact that $|\sqrt{x} - \sqrt{y}| \leq \sqrt{|x - y|}$ holds for any $x, y \geq 0$ and $\|\cdot\|_F$ denotes the Frobenius norm.

Let $\mathcal{C}_{\boldsymbol{w}}$ be an $\varepsilon/4$-cover of $\{\boldsymbol{w} \in \mathbb{R}^d : \|\boldsymbol{w}\| \leq L\}$ with respect to the L2-norm, $\mathcal{C}_{\boldsymbol{A}}$ an $\varepsilon^2/4$-cover of $\{\boldsymbol{A} \in \mathbb{R}^{d \times d} : \|\boldsymbol{A}\|_F \leq d^{1/2}B^2\lambda^{-1}\}$ with respect to the Frobenius norm, and $\mathcal{C}_v$ an $\varepsilon/2$-cover of the interval $[0, D]$. Then, treating the matrix $\boldsymbol{A} \in \mathbb{R}^{d \times d}$ as a long vector of dimension $d \times d$, and applying Lemma D.1, we know that we can find such covers with

$$\log |\mathcal{C}_{\boldsymbol{w}}| \leq d \log(1 + 8L/\varepsilon), \quad \log |\mathcal{C}_{\boldsymbol{A}}| \leq d^2 \log(1 + 8d^{1/2}B^2/(\lambda\varepsilon^2)), \quad \log |\mathcal{C}_v| \leq \log(1 + 8D/\varepsilon).$$

Hence, the set of functions

$$\mathcal{C}_Q = \{Q \in \mathbb{R}^{\mathcal{S} \times \mathcal{A}} : Q(\cdot, \cdot) = \boldsymbol{w}^T \boldsymbol{\varphi}(\cdot, \cdot) + v + \sqrt{\boldsymbol{\varphi}(\cdot, \cdot)^T \boldsymbol{A} \boldsymbol{\varphi}(\cdot, \cdot)}, \boldsymbol{w} \in \mathcal{C}_{\boldsymbol{w}}, \boldsymbol{A} \in \mathcal{C}_{\boldsymbol{A}}, v \in \mathcal{C}_v\}$$

has cardinality bounded by $\log |\mathcal{C}_Q| \leq d \log(1 + 8L/\varepsilon) + d^2 \log(1 + 8d^{1/2}B^2/(\lambda\varepsilon^2)) + \log(1 + 8D/\varepsilon)$. We can show that $\mathcal{C}_Q$ defined above is an $\varepsilon$-cover for $\mathcal{Q}$ as follows. Fix any $Q \in \mathcal{Q}$ parameterized by $(\boldsymbol{w}, v, \boldsymbol{A})$ and consider $\widetilde{Q} \in \mathcal{Q}$ parameterized by $(\widetilde{\boldsymbol{w}}, \widetilde{v}, \widetilde{\boldsymbol{A}})$ where $\widetilde{\boldsymbol{w}} \in \mathcal{C}_{\boldsymbol{w}}$ with $\|\boldsymbol{w} - \widetilde{\boldsymbol{w}}\|_2 \leq \varepsilon/4$, $\widetilde{v} \in \mathcal{C}_v$ with $|v - \widetilde{v}| \leq \varepsilon/4$ and $\widetilde{\boldsymbol{A}} \in \mathcal{C}_{\boldsymbol{A}}$ with $\|\boldsymbol{A} - \widetilde{\boldsymbol{A}}\|_F \leq \varepsilon^2/4$. Then, by the bound (3), we have $\|Q - \widetilde{Q}\|_\infty \leq \varepsilon$ as desired. This concludes the proof. $\square$

**Lemma D.3.** *Let $\mathcal{V}$ be a class of functions mapping from $\mathcal{S}$ to $\mathbb{R}$ defined as*

$$\mathcal{V} = \{\max_a Q(\cdot, a) : Q(\cdot, \cdot) = \text{CLIP}(Q_1(\cdot, \cdot); Q_2(\cdot, \cdot) \vee Q_3(\cdot, \cdot), Q_4(\cdot, \cdot) \wedge Q_5(\cdot, \cdot), \ Q_1, \ldots, Q_5 \in \mathcal{Q}\}$$

*where the function class $\mathcal{Q}$ is defined in Lemma D.2. Then,*

$$\log \mathcal{N}_\epsilon(\mathcal{V}, \|\cdot\|_\infty) \leq 5d \log(1 + 8L/\varepsilon) + 5 \log(1 + 8D/\varepsilon) + 5d^2 \log[1 + 8d^{1/2}B^2/(\lambda\varepsilon^2)].$$

*Proof.* Let $\mathcal{W}$ be a class of functions mapping from $\mathcal{S} \times \mathcal{A} \to \mathbb{R}$ of the form

$$Q(\cdot, \cdot) = \text{CLIP}(Q_1(\cdot, \cdot); Q_2(\cdot, \cdot) \vee Q_3(\cdot, \cdot), Q_4(\cdot, \cdot) \wedge Q_5(\cdot, \cdot))$$

where $Q_1, \ldots, Q_5 \in \mathcal{Q}$. Let $\mathcal{C}_0$ be an $\epsilon$-cover of the function class $\mathcal{Q}$ with size $\log |\mathcal{C}_0| \leq d \log(1 + 8L/\varepsilon) + \log(1 + 4D/\varepsilon) + d^2 \log[1 + 8d^{1/2}B^2/(\lambda\varepsilon^2)]$. Such a cover exists by Lemma D.2. Let $\mathcal{C}$ be defined as

$$\mathcal{C} = \{Q \in \mathbb{R}^{\mathcal{S} \times \mathcal{A}} : Q(\cdot, \cdot) = \text{CLIP}(Q_1(\cdot, \cdot); Q_2(\cdot, \cdot) \vee Q_3(\cdot, \cdot), Q_4(\cdot, \cdot) \wedge Q_5(\cdot, \cdot), \ Q_1, \ldots, Q_5 \in \mathcal{C}_0\}.$$

---

**Algorithm 4** $\gamma$-LSCVI-UCB (flawed initial version in arXiv by Hong et al. (2025))

---

**Input:** Discounting factor $\gamma \in [0, 1)$, regularization constant $\lambda > 0$, span $H > 0$, bonus factor $\beta > 0$.
**Initialize:** $k \leftarrow 1, t_k \leftarrow 1, \Lambda_1 \leftarrow \lambda I, Q_1(\cdot, \cdot), V_1(\cdot) \leftarrow \frac{1}{1-\gamma}$ for $t \in [T]$.
 1: Receive state $s_1$.
 2: **for** time step $t = 1, \ldots, T$ **do**
 3:   Take action $a_t = \text{argmax}_a \max_{\tau \in [t_k:t]} Q_\tau(s_t, a)$; Receive reward $r(s_t, a_t)$; Receive next state $s_{t+1}$.
 4:   $\Lambda_t \leftarrow \Lambda_{t-1} + \boldsymbol{\varphi}(s_t, a_t)\boldsymbol{\varphi}(s_t, a_t)^\top$.
 5:   $Q_{t+1}(\cdot, \cdot) \leftarrow \left( r(\cdot, \cdot) + \gamma([\widehat{P}_{t_k} V_t](\cdot, \cdot) + \beta\|\boldsymbol{\varphi}(\cdot, \cdot)\|_{\Lambda_{t_k}^{-1}}) \right) \wedge \frac{1}{1-\gamma}$.
 6:   $\widetilde{V}_{t+1}(\cdot) \leftarrow \max_a Q_{t+1}(\cdot, a)$.
 7:   $V_{t+1}(\cdot) \leftarrow \text{CLIP}(\widetilde{V}_{t+1}(\cdot); \min_{s' \in \mathcal{S}} \widetilde{V}_{t+1}(s'), \min_{s' \in \mathcal{S}} \widetilde{V}_{t+1}(s') + H)$.
 8:   **if** $2\det(\Lambda_{t_k}) < \det(\Lambda_t)$ **then**
 9:     $k \leftarrow k + 1, t_k \leftarrow t + 1$.
10:   **end if**
11: **end for**

---

Then, we have $\log|\mathcal{C}| \leq 5\log|\mathcal{C}_0|$, and we can show that $\mathcal{C}$ is an $\varepsilon$-cover of $\mathcal{W}$ as follows. Consider a function $W \in \mathcal{W}$, with $W(\cdot, \cdot) = \text{CLIP}(Q_1(\cdot, \cdot); Q_2(\cdot, \cdot) \vee Q_3(\cdot, \cdot), Q_4(\cdot, \cdot) \wedge Q_5(\cdot, \cdot))$ where $Q_1, \ldots, Q_5 \in \mathcal{Q}$. Let $\widetilde{Q}_i \in \mathcal{C}_0$ be the approximation of $Q_i$ for $i = 1, \ldots, 5$ such that $\|\widetilde{Q}_i - Q_i\|_\infty \leq \varepsilon$. Such a $\widetilde{Q}_i$ exists since $\mathcal{C}_0$ is an $\varepsilon$-cover of $\mathcal{Q}$. Let $\widetilde{Q}(\cdot, \cdot) = \text{CLIP}(\widetilde{Q}_1(\cdot, \cdot); \widetilde{Q}_2(\cdot, \cdot) \vee \widetilde{Q}_3(\cdot, \cdot), \widetilde{Q}_4(\cdot, \cdot) \wedge Q_5(\cdot, \cdot))$. Then, $\widetilde{Q} \in \mathcal{C}$ and

$$
\begin{aligned}
Q(\cdot, \cdot) &= \text{CLIP}(Q_1(\cdot, \cdot); Q_2(\cdot, \cdot) \vee Q_3(\cdot, \cdot), Q_4(\cdot, \cdot) \wedge Q_5(\cdot, \cdot)) \\
&\leq \text{CLIP}(\widetilde{Q}_1(\cdot, \cdot) + \varepsilon; (\widetilde{Q}_2(\cdot, \cdot) + \varepsilon) \vee (\widetilde{Q}_3(\cdot, \cdot) + \varepsilon), (\widetilde{Q}_4(\cdot, \cdot) + \varepsilon) \wedge (\widetilde{Q}_5(\cdot, \cdot) + \varepsilon)) \\
&= \text{CILP}(\widetilde{Q}_1(\cdot, \cdot); \widetilde{Q}_2(\cdot, \cdot) \vee \widetilde{Q}_3(\cdot, \cdot), \widetilde{Q}_4(\cdot, \cdot) \wedge \widetilde{Q}_5(\cdot, \cdot)) + \varepsilon \\
&= \widetilde{Q}(\cdot, \cdot) + \varepsilon.
\end{aligned}
$$

Similarly, we have

$$
\begin{aligned}
Q(\cdot, \cdot) &= \text{CLIP}(Q_1(\cdot, \cdot); Q_2(\cdot, \cdot) \vee Q_3(\cdot, \cdot), Q_4(\cdot, \cdot) \wedge Q_5(\cdot, \cdot)) \\
&\geq \text{CLIP}(\widetilde{Q}_1(\cdot, \cdot) - \varepsilon; (\widetilde{Q}_2(\cdot, \cdot) - \varepsilon) \vee (\widetilde{Q}_3(\cdot, \cdot) - \varepsilon), (\widetilde{Q}_4(\cdot, \cdot) - \varepsilon) \wedge (\widetilde{Q}_5(\cdot, \cdot) - \varepsilon)) \\
&= \text{CILP}(\widetilde{Q}_1(\cdot, \cdot); \widetilde{Q}_2(\cdot, \cdot) \vee \widetilde{Q}_3(\cdot, \cdot), \widetilde{Q}_4(\cdot, \cdot) \wedge \widetilde{Q}_5(\cdot, \cdot)) - \varepsilon \\
&= \widetilde{Q}(\cdot, \cdot) - \varepsilon,
\end{aligned}
$$

which shows $\|Q - \widetilde{Q}\|_\infty \leq \varepsilon$, and that $\mathcal{C}$ is an $\varepsilon$-cover of $\mathcal{W}$. Since $\max_a$ is a contraction map, it follows that $\mathcal{V} = \{\max_a Q(\cdot, a) : Q \in \mathcal{W}\}$ is covered by $\widetilde{\mathcal{V}} = \{\max_a Q(\cdot, a) : Q \in \mathcal{C}\}$. The proof is complete by observing that $\log|\widetilde{\mathcal{V}}| \leq \log|\mathcal{C}| \leq 5\log|\mathcal{C}_0|$, and that there exists $\varepsilon$-cover $\mathcal{C}_0$ for $\mathcal{Q}$ with $\log|\mathcal{C}_0| \leq d\log(1 + 8L/\varepsilon) + \log(1 + 8D/\varepsilon) + d^2\log[1 + 8d^{1/2}B^2/(\lambda\varepsilon^2)]$ by Lemma D.2. $\qquad\square$

## E. More Discussion on Previous Work

In this section, we provide a more detailed comparison with prior work. The idea of using value iteration for infinite-horizon average-reward linear MDPs via approximation from the discounted setting was first introduced by Hong et al. (2025). However, the initial arXiv version of their work contained a flaw in the analysis. This issue was later corrected in the published version, which involved significant modifications to the algorithmic structure, resulting in a substantial departure from the original version. The corrected algorithm is discussed in Section 2.4 of the main paper. For completeness, we review the original (incorrect) version in this section and explain how Hong et al. (2025) addressed the issue.

The incorrect version is shown in Algorithm 4. Unlike the corrected version (Algorithm 1), which plans for future policies for the remaining time steps by performing backward value iteration to compute action-value functions $Q_t$ for the remaining time steps, the flawed version updates the action-value function *in place*, meaning it performs a value iteration and updates the action-value function directly at each time step, and selects an action based on this updated value function. For a technical reason, the algorithm runs in episodes, starting a new episode when the covariance $\Lambda_t$ doubles (Line 8).

Although simpler and more memory efficient, the in-place action-value update scheme alone appears insufficient, and the incorrect version of the paper introduces a *max-pooling* step (Line 3) that pools $Q$-functions such that the pooled $Q$ function $\bar{Q}_t = \max_{\tau \in [t_k:t]}$ is monotonically increasing within the episode $k$. To motivate this modification, consider what happens if the action $a_t$ is directly taken with respect to the value function $Q_t$, without applying max-pooling. Note that by uniform concentration bound on $\widehat{P}_t V$, we have

$$Q_{\tau+1}(s,a) \leq r(s,a) + \gamma[PV_\tau](s,a) + 2\beta\|\varphi(s,a)\|_{\Lambda_\tau^{-1}} \tag{4}$$

for all $\tau \in [T]$, $s \in \mathcal{S}$ and $a \in \mathcal{A}$. Using the inequality with $\tau \leftarrow t+2$, $s \leftarrow s_t$, $a \leftarrow a_t$, we can bound the regret as:

$$
\begin{aligned}
R_T &= \sum_{t=1}^{T}(J^* - r(s_t, a_t)) \\
&\leq \sum_{t=1}^{T}(J^* - Q_{t+2}(s_t, a_t) + \gamma[PV_{t+1}](s_t, a_t) + 2\beta\|\varphi(s_t, a_t)\|_{\Lambda_t^{-1}} \\
&\leq \sum_{t=1}^{T}(J^* - (1-\gamma)V_{t+1}(s_{t+1})) + \sum_{t=1}^{T}(Q_{t+1}(s_{t+1}, a_{t+1}) - Q_{t+2}(s_t, a_t)) \\
&\quad + \gamma\sum_{t=1}^{T}([PV_{t+1}](s_t, a_t) - V_{t+1}(s_{t+1})) + 2\beta\sum_{t=1}^{T}\|\varphi(s_t, a_t)\|_{\Lambda_t^{-1}}.
\end{aligned}
$$

Here, the in-place nature of value iteration causes a misalignment in the second: the $Q$-function in the second part of the term is one iteration ahead of the $Q$-function in the first part. This mismatch complicates the regret analysis and undermines the telescoping cancellation typically leveraged in such proofs.

One potential remedy is to enforce monotonicity in the $Q$-function within an episode by adding a line $Q_{t+1} \leftarrow Q_t \vee Q_{t+1}$ after Line 5. With such a modification, telescoping goes through. However, this approach introduces a complication: as the $Q$-function becomes increasingly complex with each update, the covering number of the function class containing all such $Q$-functions grows exponentially with $T$. As a result, the uniform concentration bound for $\widehat{P}_t V$ becomes vacuous.

To sidestep the covering issue, the initial (incorrect) version of Hong et al. (2025) does not enforce monotonicity in $Q_t$, maintaining $Q_t$ in a function class with a low covering number. Instead, it *max-pools* the $Q$-functions as $\bar{Q}_t = \max_{\tau \in [t_k:t]} Q_\tau(s_t, a)$ and selects a greedy action $a_t$ with respect to the pooled function $\bar{Q}_t$. The idea is to make $\bar{Q}_t$ monotonically increasing in $t$ without running into covering issue. Denoting by $\tau_t(s,a) = \operatorname{argmax}_{t \in [\tau_k:t]} Q_\tau(s,a)$ for $t$ in episode $k$, and using the inequality (4) with $\tau \leftarrow \tau_t(s,a) - 1$, $s \leftarrow s_t$, $a \leftarrow a_t$, and using $Q_{\tau_t(s_t, a_t)}(s_t, a_t) = \bar{Q}_t(s_t, a_t)$, we get

$$
\begin{aligned}
R_T &\leq \sum_{t=1}^{T}(J^* - Q_{\tau_t(s_t,a_t)}(s_t, a_t) + \gamma[PV_{\tau_t(s_t,a_t)-1}](s_t, a_t) + 2\beta\|\varphi(s_t, a_t)\|_{\Lambda_t^{-1}} \\
&= \sum_{t=1}^{T}(J^* - (1-\gamma)V_{\tau_t(s_t,a_t)-1}(s_{t+1})) + \sum_{t=1}^{T}(V_{\tau_t(s_t,a_t)-1}(s_{t+1}) - \bar{Q}_t(s_t, a_t)) \\
&\quad + \gamma\sum_{t=1}^{T}([PV_{t+1}](s_t, a_t) - V_{t+1}(s_{t+1})) + 2\beta\sum_{t=1}^{T}\|\varphi(s_t, a_t)\|_{\Lambda_t^{-1}}.
\end{aligned}
$$

The incorrect version of Hong et al. (2025) bounds the second term as follows (we slightly adapt their argument for clarity).

$$
\begin{aligned}
V_{\tau_t(s_t,a_t)-1}(s_{t+1}) &\leq \widetilde{V}_{\tau_t(s_t,a_t)-1}(s_{t+1}) \\
&= \max_a Q_{\tau_t(s_t,a_t)-1}(s_{t+1}, a) \\
&\leq \max_a \max_{\tau \in [t_k:t+1]} Q_\tau(s_{t+1}, a) \\
&= \max_a \bar{Q}_{t+1}(s_{t+1}, a) \\
&= \bar{Q}_{t+1}(s_{t+1}, a_{t+1}),
\end{aligned}
$$

which allows for a telescoping sum. However, the second inequality above is wrong. Although $\tau_t(s_t, a_t) \in [t_k : t]$, $\tau_t(s_t, a_t) - 1$ is not in the interval $[t_k : t+1]$ when $\tau_t(s_t, a_t) = t_k$. Due to this *off-by-one* error in their analysis, the regret bound fails.

The corrected version fixes the issue by completely restructuring the algorithm. Specifically, it uses a backward value iteration scheme, so that the value function $Q_{t+1}$ used for selecting an action $a_{t+1}$ at time $t+1$ is one value iteration *behind* $Q_t$. With this scheme, the time index in (4) changes to:

$$Q_\tau(s, a) \le r(s, a) + \gamma[PV_{\tau+1}](s, a) + 2\beta\|\varphi(s, a)\|_{\Lambda_\tau^{-1}},$$

and the regret bound for the corrected algorithm becomes

$$\sum_t (J^* - r(s_t, a_t)) \le \sum_t (J^* - Q_t(s_t, a_t) + \gamma[PV_{t+1}](s_t, a_t) + 2\beta\|\varphi(s_t, a_t)\|_{\Lambda_t^{-1}}$$
$$\le \sum_t (J^* - (1-\gamma)V_{t+1}(s_{t+1})) + \sum_t (Q_{t+1}(s_{t+1}, a_{t+1}) - Q_t(s_t, a_t))$$
$$+ \gamma \sum_t ([PV_{t+1}](s_t, a_t) - V_{t+1}(s_{t+1})) + 2\beta \sum_t \|\varphi(s_t, a_t)\|_{\Lambda_t^{-1}}.$$

Notice the change in the time index in the second term. Without the need for further modification of the algorithm, the second term can be bounded using telescoping sum, as is done in Appendix C.3.

