# OpenReview forum: "A Computationally Efficient Algorithm for Infinite-Horizon Average-Reward Linear MDPs"
_ICML.cc/2025/Conference — ICML 2025 poster_

### Official Review · Reviewer_3Auz · 2025-03-04

**Overall Recommendation:** 4

**Summary:**

This paper tackles the open problem of proving a rate optimal regret bound in average reward infinite horizon linear mdp with a computationally efficient algorithm. The problem it is solved successfully, using a double loop structure inspired by hong et al. 2025 and a deviation controlled mechanism to make sure that the iterates produced by least square value iteration with different clipping thresholds could be bound by the difference of the clipping thresholds.

### After rebuttal

 My opinion remains positive. The assumption on bounded features and weights is clear now but I think it would be better if the authors make clear that finding features with the desired boundness properties requires computing a MVEE.

Please make sure to add the comparison with the failed attempt in Hong et al. 2025v1. I think this is very important for the placement of the paper in the literature.

**Claims And Evidence:**

Claims are convincing

**Essential References Not Discussed:**

None

**Experimental Designs Or Analyses:**

see above

**Methods And Evaluation Criteria:**

Rigorous proofs.

**Other Comments Or Suggestions:**

Make clear that Algorithm 1 and 2 need to know both $T$ and $H$, especially it seems very difficult to obtain an algorithm that works without knowing the value of $T$ in advance given the double loop structure.
The author should make this limitation clearer.

**Other Strengths And Weaknesses:**

Well written paper

**Questions For Authors:**

How does the Bellman optimality equation in Assumption A relates to structural assumptions often imposed in Average Reward infinite horizon problems such as Communicating/Weakly Communicating MDPs ? I think this should be discussed better in the manuscript before stating Assumption A.

Why in Assumption B is reasonable to assume that the reward weights are bounded by $\sqrt{d}$. I think that in case of very badly conditioned features the reward weights might need to be larger than $\sqrt{d}$ to realize the reward function.
Let imagine to have two features $[0,1]$ and $[\epsilon, 1-\epsilon]$, now to write the reward function $r=[1,0]$ the weight norm should be as large as $1/\epsilon$. However, $\epsilon$ can be made arbitrary small so the quantity $1/\epsilon$ can become arbitrary larger that $\sqrt{2}$.

Moreover, the authors say that these assumptions are without loss of generality and refer to Wei at al. 2021 to see why this is the case.

Unfortunately Wei et al 2021 is only for the tabular setting so it does not contain information about this assumption that involves linear MDPs.

**Relation To Broader Scientific Literature:**

Yes, related to the previous attempt towards solving this problem.
This is quite well done but I have a request for the author. please find it below.

To my knowledge, there was a previous attempt in the first version of the paper uploaded in arxiv that claimed to achieve the same results
but without the double loop structure and therefore with a better time complexity.
Unfortunately, the authors of Hong et al. 2025 claimed in their v2 that the previous result contains a mistake.

I think that the mistake in Hong et al 2025 is in the maximum over the past Q functions, this algorithmic  technique makes the covering number of the state value functions space to be exponential in $T$. This is indeed avoided by the current submission that
replaces the maximum over past value functions with the double loop structure.

In my opinion, it would be helpful for the community if in the current submission, you could add a paragraph explaining the differences with Hong et al. 2025 v1 and in particular explaining why the current approach fixes the issue with the previous approach.

**Theoretical Claims:**

I checked the correctness of all the theoretical statements.
I found just a minor error at the end of page 15 in the equation at line 820 $m_{t+1}$ should be replaced with $m_t$, otherwise the authors
could not conclude that the first ferm equals $V^t_u(s)$.

---

> ### Author Rebuttal · Authors · 2025-03-28
>
> Thank you very much for your thorough review and suggestions for improving the paper. Here are our responses.
>
> **Minor error in proof**. Thank you very much for catching this. We will replace $m_{t + 1}$ with $m_t$ for the equality to hold.
>
> **Relation to previous work**. Thank you for the suggestion. We fully agree that detailed comparison with Hong 2025 v1 will benefit the community. A short explanation is that Hong 2025v1 that uses "max-pooling" of Q functions, which had an error, happened to allow computationally efficient clipping without additional algorithmic modification. Hong 2025v2, which fixes the error using double loop structure could no longer use the computational efficient clipping. Our submission introduces an algorithmic modification that allows for computationally efficient clipping under the double loop structure. We will mention the attempt made by Hong 2025v1 in Section 2.4, and add detailed comparison in the Appendix.
>
> **Knowledge of T, H**. We will make it clear to the reader that $T$ is required by adding $T$ in the list of input parameters in the algorithm box, and mentioning it in the analysis when we set $\gamma = 1 - 1 / \sqrt{T}$. We will add a remark following the Theorem statement that using *doubling-trick*, we can get away with the knowledge of $T$ at the cost of incurring additional factor of $\sqrt{\log T}$ in the regret. As for the knowledge of $H$, we will remind the reader after the Optimism lemma (Lemma 3.4) and after the main theorem statement that knowledge of an upper bound of span is required. As mentioned in our response to Reviewer CyKa, we will also mention that there is a recent work that relaxes this assumption for the tabular setting, and that we leave relaxing this assumption to future work.
>
> **Discussion on Bellman optimality equation**. Thank you for the suggestion. We will add discussion on the relation to Communicating/Weakly Communicating MDPs and Ergodic assumption. Bellman optimality equation assumption is strictly weaker than Communicating/Weakly Communicating MDP which is strictly weaker than ergodic assumption. We will mention this fact with brief reasoning and citation.
>
> **On boundedness of reward vector**.
> Note that the paper by Wei et al. 2021 "Learning infinite-horizon average-reward mdps with linear function approximation." discusses this in detail in Appendix A. (Based on your comment, we think you were referring to Wei et al. 2020. "Model-free reinforcement learning in infinite-horizon average-reward markov decision processes"). Their reasoning is as follows. Given any feature mapping $\phi$ with $\Vert \phi(\cdot, \cdot) \Vert_2 \leq 1$ and a reward function in the form $r(\cdot, \cdot) = \phi(\cdot, \cdot)^\top \theta$ such that $\vert r(\cdot, \cdot) \vert \leq 1$, there always exists an invertible matrix $A \in \mathbb{R}^{d \times d}$ such that $\Vert A \phi(\cdot, \cdot) \Vert_2 \leq 1$ and $\Vert A^{-1} \theta \Vert_2 \leq \sqrt{d}$, which enables us to work with $\widetilde\phi = A \phi$ and $\widetilde\theta = A^{-1} \theta$ instead that satisfy $\Vert \widetilde\phi \Vert_2 \leq 1$ and $\Vert \widetilde\theta \Vert_2 \leq \sqrt{d}$. They argue in their proof that such a transformation $A$ is the transformation that transforms the MVEE of the feature mappings into a unit ball. When citing their work, we will add which section of their paper to refer to (Appendix A), and briefly explain their argument.

---

> > ### Comment · Reviewer_3Auz · 2025-04-01
> >
> > Thanks a lot for your response !
> > I indeed did a mistake and I was referring to Wei et al. 2020 instead of Wei et al. 2021.
> > That point is clear now but I think it would be better if the authors make clear that finding features with the desired boundness properties requires computing a MVEE.
> >
> > Please make sure to add the comparison with the failed attempt in Hong et al. 2025v1. I think this is very important for the placement of the paper in the literature.

---

> > > ### Author Response · Authors · 2025-04-04
> > >
> > > Thank you for the suggestion. We will mention the MVEE method for the transformation for ensuring boundedness, and mention that the computation of MVEE is required.
> > >
> > > As for comparison with 2025v1. We do agree that full comparison with 2025v1, not just with 2025v2/3, is important. We will devote a section in appendix for this. Unfortunately, the comparison is too technical to describe fully in the rebuttal due to space and format constraints. Here is a brief summary:
> > >
> > > - [modified notation a bit for better comparison with our submission] Single-loop algorithm (computationally inefficient) in 2025v1 has a value iteration structure that leads to a term $V_{t - 1} (s_{t + 1}) - \widetilde{V}\_{t + 1}(s\_{t + 1})$ in the regret bound. To match the subscript for bounding this term, 2025v1 introduces "max-pooling" value functions such that $V_t$ is increasing in $t$. To avoid covering issue, they keep track of the index $\tau_t$ such that $V_{\tau_t}$ is increasing in $t$. Also, they need to restart the algorithm periodically, otherwise the value functions will always be $1 / (1 - \gamma)$ at all time steps. They employ the "doubling trick" that restarts algorithm when determinant of covariance doubles. However, they erroneously claim that $V\_{\tau_t - 1}(s_{t + 1}) \leq V\_{\tau_t + 1}(s_{t + 1})$ holds. For this to make sense, $\tau_t - 1$ should be in the current episode, enforcing $\tau_t$ to be chosen by max-pooling from the **second** time step in the episode and onward. However, taking max-pooling from the second time step will not allow bounding $V\_{\tau_t - 1} \leq V\_{\tau_t + 1}$ when $\tau_t$ is exactly the second time step of the episode, since $V\_{\tau_t - 1}$ is the function function at the first time step of the episode. Due to this "off-by-one" error, the analysis fails.
> > > - Ignoring this off-by-one error and pretending that the analysis goes through, the analysis happens to allow computationally efficient algorithm with double-loop structure trivial.
> > > - 2025v2 fixes the off-by-one error by introducing a new value iteration structure (backward induction instead of "forward induction") such that the subscript is matched to $V_{t + 1}(s_{t + 1}) - \widetilde{V}\_{t + 1}(s\_{t + 1})$ by design, so that no max-pooling is required. However, the analysis on the double-loop structure with computationally efficient algorithm fails, as is discussed in our submission in Section 3.2.
> > > - Our submission introduces the new algorithm structure using the novel deviation-controlled value iteration structure, that allows bounding the difference $V_{t + 1}^t(s_{t + 1}) - \widetilde{V}\_{t + 1}^{t + 1}(s\_{t + 1})$ by the difference in clipping thresholds, which is magically bounded using telescoping sum.

---

### Official Review · Reviewer_w193 · 2025-03-09

**Overall Recommendation:** 3

**Summary:**

This paper proposed an algorithm for infinite-horizon average-reward reinforcement learning with linear function approximation. The main problem need to be solved is the computation issue arises from minimize the value function in large state space. To address this, the paper proposed a new clipping technique, and proved that this method achieves the same regret order while enjoy computational efficiency under the large state space setting.

**Claims And Evidence:**

Yes. The paper clearly provided the computational complexity to support that the proposed algorithm runs in polynomial time.

**Essential References Not Discussed:**

N/A

**Experimental Designs Or Analyses:**

N/A

**Methods And Evaluation Criteria:**

Yes.

**Other Comments Or Suggestions:**

N/A

**Other Strengths And Weaknesses:**

Strength: The explanation of the motivation and idea of why and how to do clipping is clear and easy to understand.

Weakness: The work lack some (at least) toy example to illustrate the effectiveness of the proposed algorithm.

**Questions For Authors:**

N/A

**Relation To Broader Scientific Literature:**

N/A

**Theoretical Claims:**

I checked the proof in the main paper and appendix A, and do not find any issues.

---

> ### Author Rebuttal · Authors · 2025-03-31
>
> Thank you for your review. Here is our response.
>
> **Simulation in toy setting**. Thank you for the suggestion for running a simulation in a toy setting. We have confirmed that our algorithm runs in a *tabular setting* (i.e. linear MDP setting with feature vectors orthogonal) with average regret reasonably low when treating the bonus factor beta as a hyperparameter, as is usually done in UCB-type algorithms.
>
> To our knowledge, no study to date has conducted a simulation under the *linear MDP* setting. Nevertheless, we agree that providing a simple simulation would be valuable to the community, and we will aim to include a simulation. One reason why simulation under linear MDP is absent in the literature is due to the complication of specifying the vector of measures $\mu$ that ensures $P(s' | s, a) = \langle \mathbf\varphi(s, a), \mathbf\mu(s') \rangle$ is a valid probability measure. However, we believe we can get around the problem by restricting the feature vector $\varphi(s, a)$ to have L1-norm equal to 1, and each of the $d$ measures in $\mathbf\mu$ are probability measures. We will aim to conduct experiments under the *restricted* linear MDP setting.

---

### Official Review · Reviewer_nuyD · 2025-03-14

**Overall Recommendation:** 4

**Summary:**

This paper proposes a computationally efficient algorithm for infinite-horizon average reward linear MDPs. The paper seeks to improve upon the previously proposed approach $\gamma$-LSVI-UCB by Hong et al'25. The main contribution is that the algorithm proposed in Hong et al.'25 requires to iterate over all the state-space to find the minimum value function for clipping, which might be computationally expensive, especially for linear MDP. Hong et al.'25 assumed some minimum oracle, this paper seeks to improve upon that. The paper proposes a novel idea where one only needs to iterate over the state encountered so far. The paper in particular focuses on bounding the error $V$- $\tilde{V}$. The paper significantly contributes in this direction.

##Post Rebuttal Update##

All my concerns have been resolved, and I am happy to accept this paper.

**Claims And Evidence:**

The proofs of the claims are provided.

**Essential References Not Discussed:**

The reviewer does not have any major concern.

**Experimental Designs Or Analyses:**

N/A.

**Methods And Evaluation Criteria:**

The paper is theoretical in nature.

**Other Comments Or Suggestions:**

N/A

**Other Strengths And Weaknesses:**

Strengths:

1. The theoretical contributions are significant.

2. The paper is well-written.

Weakness:

1. The algorithm still needs to iterate over all the states encountered, which can be significant for the large time period.

2. Some simulations would be nice to see the difference in terms of computational time and the regret compared to the state-of-the-art approaches.

**Questions For Authors:**

1. Can the authors summarize briefly the main technical novelties? in particular, how did they bound the deviation?

**Relation To Broader Scientific Literature:**

The reviewer does not have any major concern.

**Theoretical Claims:**

The theoretical claims seem to be correct.

---

> ### Author Rebuttal · Authors · 2025-03-30
>
> Thank you for your time for the review and for the suggestion for improving the paper.
>
> **Q1: main technical novelties**. We will integrate the following summary in the introduction section and in Section 3.2 when introducing the method.
>
> 1. The main technical novelty lies in designing a clipped value iteration algorithm that ensures each successively generated value function deviates from its predecessor by no more than the difference between their respective clipping thresholds.
> 2. We show that a naive adaptation of previous work cannot control this deviation in the linear MDP setting.
> 3. A natural workaround would be to clip each newly generated value function with a new threshold so that it deviates from the previously generated value functions by at most the difference in the thresholds. But doing so will make the value functions more and more complex successively, running into an issue when using covering argument for uniform concentration bound.
> 4. To address this, we design a novel way for controlling the deviation without running into covering issues. Specifically, we "pool" latest value functions in such a way that the difference in successive "pooled" value functions is bounded by the the difference in the thresholds. The pooled value function has low complexity so that the function class that captures the pooled value function has low covering number.

---

> > ### Comment · Reviewer_nuyD · 2025-04-01
> >
> > I thank the authors for their responses.
> >
> > Just one clarification question. This is regarding the comment made by the reviewer 3 Auz.
> >
> > *``To my knowledge, there was a previous attempt in the first version of the paper uploaded in arxiv that claimed to achieve the same results but without the double loop structure and therefore with a better time complexity. Unfortunately, the authors of Hong et al. 2025 claimed in their v2 that the previous result contains a mistake.*
> >
> > *I think that the mistake in Hong et al 2025 is in the maximum over the past Q functions, this algorithmic technique makes the covering number of the state value functions space to be exponential in
> > . This is indeed avoided by the current submission that replaces the maximum over past value functions with the double loop structure."*
> >
> > You have responded to this comment. Can you please clarify and elaborate what the mistake was? How are you overcoming it in this version? Thanks,

---

> > > ### Author Response · Authors · 2025-04-04
> > >
> > > Here is a brief explanation regarding the comment made by reviewer 3Auz:
> > >
> > > - Hong et al. 2025v1 uses single-loop structure and uses the scheme of taking maximum of past value functions. It is known that taking maximum of past value functions in linear MDP leads to a covering issue because the function becomes more and more complex and the covering number of the function class that captures such functions become exponential in the number of time steps. As a workaround, they use a scheme of keeping track of the time index that gives the maximum value, i.e., $\tau_t(s, a) = \arg\max_{\tau \in [\tau_0:t]} Q_\tau(s, a)$ where $\tau_0$ is some reference time step (exact definition they use is a bit different). However, the analysis has an off-by-one error.
> > > - Hong et al. 2025v2 fixes the problem by introducing a double-loop structure, at the cost of incurring additional factor of $T$ in computational complexity. However, the double-loop structure requires computing the minimum of value function over the entire state space when clipping.
> > > - Our submission introduces a novel algorithm structure that allows for computationally efficient clipping.
> > >
> > > For more detailed explanation, please see our response to Reviewer 3Auz.

---

### Official Review · Reviewer_CyKa · 2025-03-15

**Overall Recommendation:** 3

**Summary:**

In this paper, the authors have studied the reinforcement learning algorithm for linear MDPs in an infinite-horizon average-reward setting. Previous works approximate the average reward by the discounted one and employ a clipping-based value iteration method. However, it requires the computation of minimum of the value function over the state space. This may be computationally prohibitive. Towards that, in this paper, an efficient clipping technique is introduced for value iteration algorithm. This requires computation of minimum value function over states visited by the algorithm. It has been established that the proposed scheme demonstrates the same regret bound as that of the previous work with a substantial drop in computational complexity which is independent of the size of the state space.
## update after rebuttal
The response addresses my major concerns. I have raised my score to 3.

**Claims And Evidence:**

Yes

**Essential References Not Discussed:**

No

**Experimental Designs Or Analyses:**

Experimental results are absent in the paper.

**Methods And Evaluation Criteria:**

Yes

**Other Comments Or Suggestions:**

Not applicable.

**Other Strengths And Weaknesses:**

The paper is well-written and contains some interesting ideas. The problem considered by the authors are relevant and interesting. However, the solution is based on some strong assumptions such as knowledge of $sp(v^*)$. Some results need clarification and lacks intuition.

**Questions For Authors:**

My comments are as follows:
1.	What if the underlying Markov chain is irreducible under various policies? In the worst case, the algorithm can still visit all the states and hence, the computational burden may still be high. I expected some simulations too in this direction to verify the efficiency of the proposed scheme.
2.	Why do we need to assume that $r\in[0,1]$? I guess removing this assumption also does not create any problem.
3.	It is assume that $sp(v^*)$ is known the learner. The authors stated that this assumption can be relaxed to the knowledge of an upper bound on $sp(v^*)$. However, in this case regret will scale with the upper bound. Knowledge of $sp(v^*)$ or an upper bound on it can both be difficult to obtain. Also, choice of a loose upper bound can result in poor regret. Is there a way in which such a requirement of knowledge can be avoided completely?
4.	Please describe what do you mean by covering number.
5.	Lemma 3.4 holds when $H\ge 2sp(v^*)$. How do one guarantee that?
6.	In Theorem 3.6, we need choose $\gamma=1-\sqrt{1/T}$. However, typically $\gamma$ is not in designer’s hand. This makes the algorithm depart from reality.
7.	It is stated that the computational complexity is independent of the size of the states space. However, $d$ is related to the size of the state space. Why is the regret large when $d$ is large? As $d$ becomes close to the size of the state space, regret should be less. Please elaborate

**Relation To Broader Scientific Literature:**

In this paper, the authors have studied the reinforcement learning algorithm for linear MDPs in an infinite-horizon average-reward setting. It has been established that the proposed scheme demonstrates the same regret bound as that of the previous work with a substantial drop in computational complexity which is independent of the size of the state space.

**Theoretical Claims:**

I have checked the proof , but not in great detail.

---

> ### Author Rebuttal · Authors · 2025-03-28
>
> Thank you for your time for the detailed review. Your feedback will help us in improving our paper. Here are our responses to your questions.
>
> **Q1: Computational issue \& Simulation**. You are correct to note that, in the large state space regime, the number of unique states visited over $T$ time steps can be very large, potentially with no repeats. However, our algorithm’s computational complexity does not scale with the size of the entire state space, since we do not enumerate all states. Instead, in the worst case, where states never repeat, the complexity scales with $T$.
> Regarding simulation, due to the theoretical nature of the linear MDP assumption, designing a concrete simulation environment is challenging and, to our knowledge, no simulation has been performed in the linear MDP setting in the literature. we agree that providing a simple simulation would be valuable to the community, and we will aim to include a simulation under a *restricted* linear MDP setting described in our response to Reviewer w193.
>
> **Q2: Why assume reward to be bounded**. Your intuition that the assumption can be removed is correct. And in fact, it is this intuition that It is standard in the literature to assume $r \in [0, 1]$ without loss of generality. When the actual reward lines in $[0, B]$, say, then we can scale it to $\widetilde{r} \in [0, 1]$, analyze the regret bound under the new reward, and then multiply the final regret bound by $B$ to account for the scaling. To avoid carrying the constant $B$ around in our derivations, we assumed $r \in [0, 1]$. We will add a clarifying note on this point in our final version of the paper.
>
> **Q3, Q5: Assumption on the knowledge of span**. Thank you for bringing this up. Indeed, requiring prior knowledge of $\text{sp}(v^\ast)$ is a limitation of our approach. Whether sample-efficient learning is possible without knowing the span in advance has been an open question for a long time, and many existing works rely on this assumption. A recent result by Boone et al. [1] shows for the first time that it can be avoided in the tabular setting. However, extending their technique to the linear MDP setting is non-trivial and will likely require a major breakthrough. We leave this to future work. We will add a remark on this assumption and cite [1] in the final version of the paper.
>
> **Q4: Covering number**. Thank you for the suggestion. In the paragraph following Lemma 3.3, we will describe the general covering argument that uses epslion-net for covering the function class for getting uniform concentration bound and briefly describe $\epsilon$-covering number of a function class and how the log covering number scales the concentration bound.
>
> **Q6: Setting discounting factor**. The original problem the designer is trying to solve is the average-reward setting. We are proposing to approximate the average-reward setting by a discounted setting with a particular discounting factor. The designer will work with the discounted setting with discounting factor set to the proposed value. So, it is in the hand of the designer to set the discounting factor. We will make it clear in the beginning of Section 3 that the discounting factor is set by the designer to approximate the average-reward setting.
>
> **Q7: Size of the state space and the dimension**. Thank you for the question. When we say "size of the state space", we mean the number of the states in the state space. In the tabular setting, we do not assume any structure in the MDP that allows generalizing to unseen states, and the computational and statistical complexity both scale with the number of states. However, in the linear MDP setting, we assume a structure in the MDP that allows for the generalization through the feature mapping $\varphi$ that maps state-action pairs to a $d$-dimensional feature vector. Due to the feature representation, conceptually, we no longer need to learn about all the states individually. Rather, we only need to learn about $d$ "directions", which allows for computational and statistical efficiency. Specifically, the computational complexity and the regret bound scales with $d$, not the size of the state space. With this intuition, we can see that if $d$ is large, then there is more "directions" to explore, making both computational complexity and regret large. We will add a summary of this discussion when defining the linear MDP setting in Section 2.3.
>
> ---
>
> [1] Achieving Tractable Minimax Optimal Regret in Average Reward MDPs. NeurIPS 2024.

---

> > ### Comment · Reviewer_CyKa · 2025-04-05
> >
> > I thank the authors for their detailed response. However, I am still concerned regarding two aspects:
> > 1.	Knowledge of $sp(v^*)$: Can this assumption be relaxed?
> > 2.	The authors stated that when $d$ is large, then there is more directions to explore, making both computational complexity and regret large. It is not clear to me. Please explain using two extreme cases, $d=1$ and $d=|\mathcal{S}\times \mathcal{A}|$? Since there is no approximation involved in the second case, won’t the regret due to approximation be less in the second case?
> >
> > Another clarification question: Since the original problem considers an infinite horizon average-reward setting, how do you choose $T$ to set $\gamma=1-\sqrt{1/T}$? Will any choice of large $T$ be fine to represent the infinite horizon setting?

---

> > > ### Author Response · Authors · 2025-04-07
> > >
> > > **Q1** Relaxing the assumption of the knowledge of span in linear MDPs is an open problem. The authors of "Achieving Tractable Minimax Optimal Regret in Average Reward MDPs, NeurIPS 2024" relaxes the assumption in tabular setting by using a subroutine that estimates the optimal bias function $v^\ast$ to work without the knowledge of $\text{sp}(v^\ast)$. Their approach is based on the observation that, when an optimal policy is run, the bias difference $v^\ast(s) - v^\ast(s')$ is roughly the difference in values when starting from the state $s$ and $s'$. They use the average reward collected in subtrajectories that start at $s$ and end at $s'$ to estimate $v^\ast(s) - v^\ast(s')$. It is unclear how to generalize this idea to (linear) function approximation setting where the state space is large, since their idea relies on tracking each pair of $(s, s')$ to estimate $v^\ast(s) - v^\ast(s')$, which can be sample inefficient in function approximation setting.
> > >
> > > Designing an algorithm (even one that is computationally inefficient) for linear MDP without the knowledge of the span is an interesting and challenging open problem, and would be a good topic for a future standalone paper.
> > >
> > > **Q2** In general, in the learning setting where the transition probability $P$ is unknown, the algorithm design centers around estimating the unknown $P$ or a quantity that depends on the unknown $P$. Typically, in value-iteration based algorithm, which is the algorithm class we use, the quantity of interest is $[PV](s, a)$. The regret bound depends on how sample efficiently we can estimate $[PV](s, a)$ since the regret bound scales with the width of the confidence bound. (in general, regret of UCB type of algorithm scales with sum of confidence bounds).
> > > In linear MDP setting, we exploit the fact that $[PV](s, a) = \langle \varphi(s, a), w \rangle$ for some $w \in \mathbb{R}^d$, and use linear regression to estimate $w$, which gives the concentration bound that scales linearly with $d$ (see Lemma 3.3). This suggests that the regret will scale with $d$. To gain more intuition, consider the following two extreme cases.
> > >
> > > - $d = 1$. In this case, the transition $P(\cdot | s, a)$ is the same for all $s, a$ pairs. Hence, we don't have to collect transition data for all $(s, a)$ pairs separately to be able to estimate $P(\cdot | s, a)$ for each $s, a$. We can "pool" data for all pairs of $(s, a)$ to estimate $P(\cdot | s, a)$. Similarly, estimation of $[PV](s, a)$ can be done by sample average of $V(s')$.
> > > - $d = SA$. In this case the transition $P(\cdot | s, a)$ can be arbitrary for each $(s, a)$ pair. Hence, unlike the $d = 1$ case, we cannot "pool" data, and requires data for each $(s, a)$ pair. Similarly, estimating $[PV](s, a)$ requires data for each $(s, a)$ pair. Intuitively, this leads to a concentration bound that scales with $SA$ and hence regret scales with $SA$.
> > >
> > > We will make it clearer that the concentration bound in Lemma 3.3 scales with $d$, and that the concentration bound eventually enters the regret bound.
> > >
> > > **Regarding $T$**. We first clarify the distinction between the problem setting ("infinite-horizon") and the performance guarantee of an algorithm ("finite-time regret bound"). The notion "infinite-horizon average-reward" describes the problem setting where the criterion for evaluating a policy is through the infinite-horizon average-reward. When analyzing the performance of an algorithm, one typically uses "finite-time regret bound" with a certain time $T$ and show a bound on the $T$-step where regret in each step is against the optimal average-reward.
> > > Our guarantee allows a designer to choose $\gamma = 1 - 1 / \sqrt{T}$ to achieve $O(\sqrt{T})$ $T$-step regret.
> > >
> > > As you hinted, a designer may want a single algorithm that guarantees $O(\sqrt{t})$ $t$-step regret for *any* $t$. Algorithms with this guarantee is called **anytime algorithms**. There is a known reduction called "doubling trick" that uses algorithms with $O(\sqrt{T})$ $T$-step regret to design an anytime algorithm. The reduction is as follows.
> > >
> > > Suppose we have access to an algorithm $Alg(T)$ that guarantees $O(\sqrt{T})$ $T$-step regret for each $T$. In our paper, $Alg(T)$ can be obtained by choosing $\gamma = 1 - 1 / \sqrt{T}$. Then, we can design an anytime algorithm by running $Alg(2^n)$ for $2^n$ steps for $n = 0, 1, 2, \dots$. That is, run $Alg(1)$ for 1 time step, then run $Alg(2)$ for 2 time steps, then run $Alg(4)$ for 4 time steps, etc. For any $2^{n - 1} \leq t < 2^n$, such an algorithm guarantees $t$-step regret bound of
> > >
> > > $$
> > > O(\sqrt{1}) + O(\sqrt{2}) + \cdots + O(\sqrt{2^n}) = O(\sqrt{n 2^n}) = O(\sqrt{t \log t})
> > > $$
> > > by Cauchy-Schwarz, since $1 + 2 + ... + 2^n \approx O(2^n)$.
> > >
> > > We realize that discussing anytime algorithm would greatly improve clarity on our guarantee. We will incorporate this discussion into the paper. Also, we think parameterizing the algorithm with $T$ instead of $\gamma$ may be clearer for this discussion.

---

### Decision · Program_Chairs · 2025-05-01

**Decision:**

Accept (poster)

**Comment:**

The paper studies reinforcement learning in infinite-horizon average-reward settings with linear MDPs. Comparing to the prior work, this work's algorithm relaxes the condition and only requires the algorithm to know the minimum of the value functions over the set of the states visited by the algorithm instead of the entire state space. Thus the work improves prior work in terms of computation complexity.

Reviewers in general are positive about the paper and the authors rebutal also successfully addressed some of the concerns from the reviewers.

I think some suggestions from the reviewers are indeed very good and can make the paper stronger. Please make sure to include the following discussions in the paper: (1) discussion about the comparison to the previous work from Hong et al. 2025v1, and discuss the issues related to the covering number and how the techiniques in this paper handles this issue; (2) discuss how to ensure the boundness assumptions using MVEE and the computation cost from MVEE. I think adding these discussions would make the paper more complete.